# Therapeutic resistance in acute myeloid leukemia cells is mediated by a novel ATM/mTOR pathway regulating oxidative phosphorylation

Hae J Park[1,2], Mark A Gregory[1], Vadym Zaberezhnyy[1], Andrew Goodspeed[3,4], Craig T Jordan[5], Jeffrey S Kieft[1], James DeGregori[1,4,5]*

[1]Department of Biochemistry and Molecular Genetics, School of Medicine, University of Colorado Anschutz Medical Campus, Aurora, United States; [2]Medical Scientist Training Program, University of Colorado Anschutz Medical Campus, Aurora, United States; [3]Department of Pharmacology, University of Colorado Anschutz Medical Campus, Aurora, United States; [4]University of Colorado Comprehensive Cancer Center, University of Colorado Anschutz Medical Campus, Aurora, United States; [5]Department of Medicine, Section of Hematology, University of Colorado Anschutz Medical Campus, Aurora, United States

*For correspondence:
James.Degregori@cuanschutz.edu

Competing interest: The authors declare that no competing interests exist.

**Abstract** While leukemic cells are susceptible to various therapeutic insults, residence in the bone marrow microenvironment typically confers protection from a wide range of drugs. Thus, understanding the unique molecular changes elicited by the marrow is of critical importance toward improving therapeutic outcomes. In this study, we demonstrate that aberrant activation of oxidative phosphorylation serves to induce therapeutic resistance in FLT3 mutant human AML cells challenged with FLT3 inhibitor drugs. Importantly, our findings show that AML cells are protected from apoptosis following FLT3 inhibition due to marrow-mediated activation of ATM, which in turn upregulates oxidative phosphorylation via mTOR signaling. mTOR is required for the bone marrow stroma-dependent maintenance of protein translation, with selective polysome enrichment of oxidative phosphorylation transcripts, despite FLT3 inhibition. To investigate the therapeutic significance of this finding, we tested the mTOR inhibitor everolimus in combination with the FLT3 inhibitor quizartinib in primary human AML xenograft models. While marrow resident AML cells were highly resistant to quizartinib alone, the addition of everolimus induced profound reduction in tumor burden and prevented relapse. Taken together, these data provide a novel mechanistic understanding of marrow-based therapeutic resistance and a promising strategy for improved treatment of FLT3 mutant AML patients.

## Editor's evaluation

FLT3 (Fms-related receptor tyrosine kinase 3) activation occurs in a subset of AML cases and is associated with poor prognosis. This work is focused on the mechanisms of resistance to FLT3 inhibitors in AML. The authors show that the combination of the FLT3 inhibitor and an mTORC1 inhibitor reduces tumor burden and prevents relapse in FLT3 mutant AML. This article is of interest to scientists and physicians investigating AML as well as scientists studying signaling pathways.

## Introduction

Acute myeloid leukemia (AML) is an aggressive blood cancer with a high relapse rate and resistance to cytotoxic therapies (*Brumatti et al., 2017*). Internal tandem duplication (ITD) mutations in FMS-like tyrosine kinase 3 (FLT3) are among the most prevalent mutations in AML and are particularly associated with a poor prognosis (*Burnett et al., 2011*). FLT3-ITD leads to constitutive activation of FLT3, a receptor tyrosine kinase, which induces activation of multiple effector molecules involved in survival, proliferation, and cell growth (*Fathi and Chen, 2011*; *Gilliland and Griffin, 2002*). Clear scientific and clinical evidence that supports the significance of activated FLT3 in leukemogenesis (*Smith et al., 2012*) has led to the development of several FLT3-targeted inhibitors.

Several clinical trials with two highly potent and selective second-generation FLT3 inhibitors, quizartinib and gilteritinib, have demonstrated their effectiveness in refractory AML patients with activating FLT3 mutations, showing significantly higher overall survival and rate of remissions compared to salvage chemotherapy (*Cortes et al., 2018a*; *Cortes et al., 2018b*; *Cortes et al., 2019*; *Perl et al., 2017*; *Perl et al., 2019*). However, those remissions were short-lived. Furthermore, both FLT3 inhibitors showed noticeably delayed responses in the bone marrow (BM) (over weeks to a few months) compared to the peripheral blood (a few days), often showing persistence of the FLT3-ITD mutation in recovering marrow cells (*Levis and Perl, 2020*).

Indeed, it has long been appreciated that multiple components of the BM microenvironment promote leukemogenesis, drug resistance, and relapse (*Shafat et al., 2017*; *Tabe and Konopleva, 2015*). An extensive number of studies have shown that bone marrow stromal cells (BMSCs) provide protection to leukemia cells from chemotherapies either through direct cell-to-cell contact or secretion of soluble factors (*Bakker et al., 2016*). In particular, several recent studies have attempted to unravel the key human BMSC-activated signaling pathways in FLT3-ITD AML cells that lead to resistance to FLT3 inhibitors. In these studies, reactivation of STAT5 signaling and/or the MAPK/ERK pathway by human BMSCs was found to mediate the resistance to FLT3 inhibitors (*Patel et al., 2020*; *Yang et al., 2014*; *Traer et al., 2016*; *Sung et al., 2019*), yet conflicting data between each study indicates that the drug resistance mechanism is complex and likely to be cell-type-dependent. Thus, further studies are needed to better understand the underlying mechanism of resistance to FLT3 inhibitors.

In this study, we uncover a novel pathway mediated by ATM and mTOR whereby BMSCs protect FLT3-ITD AML cells from apoptosis following FLT3 inhibition. We demonstrate that this ATM/mTOR pathway plays an essential role in the maintenance of protein translation and oxidative phosphorylation, which is critical for cell survival. Our data show that targeting these key mediators in collaboration with FLT3 inhibitor effectively overcomes BMSC-mediated protection and enhances elimination of AML cells.

## Results

### Conditioned media of human bone marrow stromal cells prevents apoptosis in FLT3-ITD AML cells upon FLT3 inhibition

To study the paracrine effects mediated by BMSC, we utilized conditioned media from HS-5 human bone marrow stromal cells (hBMSC-CM). The HS-5 cell line has been shown to be a reliable model to reproduce the biological properties mediated by BM mesenchymal stromal cells (*Adamo et al., 2020*). Human AML cells either cultured with conditioned media from HS-5 or co-cultured with HS-5 exhibit enhanced resistance to multiple therapies, including chemotherapy, immunotherapy, and targeted therapy (*Chen et al., 2016*; *Garrido et al., 2001*; *Chen et al., 2015b*; *Edwards et al., 2019*). To further examine the role of hBMSC-CM on FLT3-ITD AML cells following FLT3 inhibition, we used MOLM-13 and MV4-11 cell lines. A significantly lower percentage of apoptotic cells were observed in both cell lines post-treatment with quizartinib in the presence of hBMSC-CM compared to regular RPMI media (*Figure 1A*). The protective effect of hBMSC-CM was also evident in the greater maintenance of viable cells observed after drug removal (*Figure 1B*). Notably, the degree of hBMSC-CM-mediated protection from apoptosis upon FLT3 inhibition was greater in MOLM-13 cells compared to MV4-11 cells. In addition, we observed similar protection by hBMSC-CM for AML cells treated with gilteritinib (*Figure 1—figure supplement 1*), another highly selective and potent FLT3 inhibitor.

Next, we measured the activity of FLT3 in cells treated with quizartinib in either regular RPMI media or hBMSC-CM. Quizartinib completely inhibited the activity of FLT3 even in the presence of

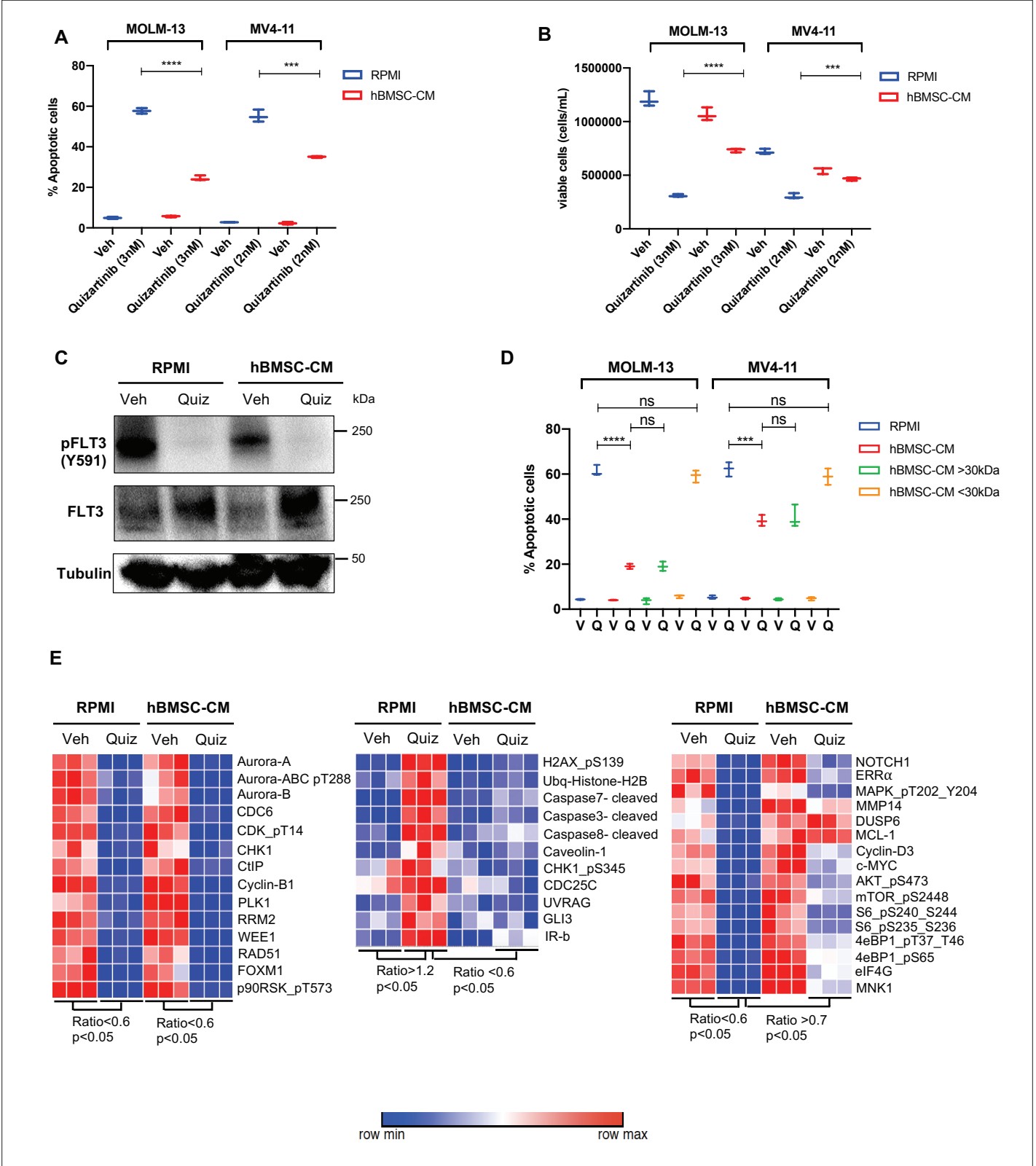

**Figure 1.** Conditioned media of human bone marrow stromal cells (hBMSCs) prevents apoptosis in FLT3-ITD acute myeloid leukemia (AML) cells upon FLT3 inhibition. (**A, B**), FLT3-ITD AML cell lines, MOLM-13 and MV4-11, were treated with either DMSO (veh) or quizartinib with indicated doses for 48 hr in regular media (RPMI) or 50% conditioned media of human BM stromal cells in RPMI (hBMSC-CM). (**A**) Apoptotic cells were detected by 7-AAD/ Annexin V staining post-treatment. (**B**) Cell viability was measured based on propidium iodide (PI) exclusion after 5 days of rebound growth post-

*Figure 1 continued on next page*

*Figure 1 continued*

treatment (n = 3). (**C**), MOLM-13 cells were harvested after 15 hr of indicated treatments (Quiz: 3 nM quizartinib) to determine FLT3 activation by Western blotting. (**D**), MOLM-13 and MV4-11 cells were treated with either vehicle (V) or quizartinib (Q) for 48 hr in RPMI, hBMSC-CM, RPMI supplemented with heavy fraction (>30 kDa) of hBMSC-CM or light fraction (<30 kDa) of hBMSC-CM, followed by detection of apoptotic cells. (**E**), MOLM-13 cells were harvested after 16 hr of treatments as indicated and subjected to reverse-phase protein array analysis to explore the expression/activation levels of a wide range of proteins.

The online version of this article includes the following source data and figure supplement(s) for figure 1:

**Source data 1.** Unedited raw blots for *Figure 1C* are shown.

**Source data 2.** Unedited blots with labels for *Figure 1C* are shown.

**Figure supplement 1.** Human bone marrow stromal cells (hBMSC-CM) provide protection against apoptosis in FLT3-ITD acute myeloid leukemia (AML) cells following FLT3 inhibition by gilteritinib.

**Figure supplement 2.** Human bone marrow stromal cell (hBMSC-CM)-mediated protection does not coincide with alterations of the cell cycle.

**Figure supplement 3.** Human bone marrow stromal cell (hBMSC-CM) from HS-27A does not provide protection against apoptosis in FLT3-ITD acute myeloid leukemia (AML) cells following FLT3 inhibition.

**Figure supplement 4.** Human bone marrow stromal cell (hBMSC-CM)-mediated protection is partially mimicked by BM-enriched cytokines.

hBMSC-CM (*Figure 1C*), suggesting that the drug resistance mediated by hBMSC-CM is not due to increased drug efflux or other mechanism preventing FLT3 inhibition. To examine whether hBMSC-CM induces differential regulation of cell cycle upon FLT3 inhibition, we next measured cell cycle profiles in FLT3-ITD AML cells upon FLT3 inhibition in either RPMI or hBMSC-CM. Our cell cycle analysis demonstrated that cells surviving FLT3 inhibition show a similar G1 phase arrest, regardless of hBMSC-CM exposure (*Figure 1—figure supplement 2*).

Given that hBMSC-CM could contain different levels of metabolites and nutrients compared to regular RPMI media, we further tested whether FLT3-ITD AML cells respond differently to quizartinib in regular RPMI supplemented with fractionated hBMSC-CM. RPMI supplemented with the heavy fraction of hBMSC-CM that contains molecules larger than 30 kDa phenocopied the protective effect of hBMSC-CM, while the light fraction of hBMSC-CM containing molecules smaller than 30 kDa demonstrated complete loss of protection from apoptosis upon FLT3 inhibition, suggesting that the protective effects of hBMSC-CM are due to larger molecules and cannot be explained by altered levels of smaller metabolites and nutrients (*Figure 1D*). Moreover, conditioned media from HS-27A, another human BM stromal cell line, did not provide protection from apoptosis upon FLT3 inhibition (*Figure 1—figure supplement 3*). Together with the previous finding that only HS-5 cells recapitulated the general expression pattern of primary BM mesenchymal stromal cells while HS-27A cells displayed a distinctive gene expression profile (*Adamo et al., 2020*), our data indicate that protection may require specific soluble factor(s). To understand the key soluble factors responsible for hBMSC-CM-mediated protection, we examined whether cytokines highly expressed by BMSCs such as GM-CSF, IL-3, and FGF-2 that have been shown to protect FLT3-ITD AML cells from cell killing by FLT3 inhibitors (*Traer et al., 2016*; *Sung et al., 2019*) show similar effects to hBMSC-CM. While GM-CSF or/and IL-3 induced modest and partial protection from apoptosis upon FLT3 inhibition, FGF-2 did not result in significant protection (*Figure 1—figure supplement 4*), indicating that hBMSC-CM-mediated survival of FLT3-ITD AML cells from FLT3 inhibition likely involves multiple factors.

Our results above indicate that the protective effect of hBMSC-CM is not due to the prevention of FLT3 inhibition or alteration of cell cycle profiles. Hence, we further focused on hBMSC-CM-induced potential intrinsic changes that may promote cell survival following FLT3 inhibition. To explore alterations in protein expression and activation in FLT3-ITD AML cells caused by both FLT3 inhibition and hBMSC-CM exposure, we pursued a nonbiased approach using reverse-phase protein array (RPPA). We observed three major patterns of change: (1) downregulation upon FLT3 inhibition in both regular RPMI and hBMSC-CM (*Figure 1E*, left panel), (2) upregulation with FLT3 inhibition in RPMI but not in hBMSC-CM (*Figure 1E*, middle panel), or (3) downregulation in RPMI but not in hBMSC (*Figure 1E*, right panel). We found that the major components of the first pattern largely consist of cell cycle regulatory proteins (AURORA kinases, CDC6, CDK, CHK1, Cyclin B1, PLK1, and WEE1), consistent with our cell cycle analysis (*Figure 1D*). Regarding the second pattern of change (upregulation by FLT3 inhibition in RPMI but not in hBMSC-CM), proteins involved with the DNA damage response (p-H2AX, ubiquitinated histone H2b, p-CHK1, CDC25C, and UVRAG) or apoptosis (cleaved caspases

3, 7, and 8) were found to be in this category, consistent with our observations of differential induction of apoptosis. Finally, proteins involved in a wide range of biological functions were found to show a pattern of downregulation with FLT3 inhibition in RPMI but significantly less downregulation or nearly complete maintenance in hBMSC-CM. This group included the oncogenic transcription factor c-MYC, antiapoptotic protein MCL-1, and MAPK/ERK signaling proteins (DUSP6 and p-MAPK/ERK). Among these, MAPK/ERK signaling (*Yang et al., 2014*) and MCL-1 (*O' Reilly et al., 2018*) are consistent with findings from other studies. Notably, multiple proteins involved in the mTOR signaling pathway (p-AKT, p-mTOR, p-S6, p-4E-BP1, eIF4G, and MNK1) demonstrated this consistent pattern. Together, the above data suggest that hBMSC-CM dramatically alters expression and activation of multiple pathways, and these changes may contribute to the survival of FLT3-ITD AML cells following FLT3 inhibition.

## Bone marrow microenvironment substantially limits downregulation of mTOR and MYC pathway in FLT3-ITD AML cells upon FLT3 inhibition in vitro and in vivo

To characterize the influence of the hBMSC-CM on gene expression in FLT3-ITD AML cells upon FLT3 inhibition, we compared the transcriptomes of MOLM-13 cells treated with either vehicle or quizartinib in RPMI or hBMSC-CM by performing RNA sequencing (RNA-seq). As shown in *Figure 2A*, FLT3 inhibited AML cells in hBMSC-CM displayed a distinct gene expression signature compared to those in RPMI. Gene set enrichment analyses (GSEA) indicated that multiple pathways related to cell fate determination are significantly more enriched in AML cells in hBMSC-CM compared with RPMI after FLT3 inhibition (*Figure 2B*). Specifically, a majority of mTOR complex1 (mTORC1) signaling genes and MYC target genes demonstrated significant suppression post-FLT3 inhibition in RPMI, while such downregulation was at least partially reversed in the presence of hBMSC-CM (*Figure 2C*), further supporting our findings from the RPPA data. We observed that hBMSC-CM induced higher base-line mTOR activity and substantially prevented downregulation of mTOR activity by FLT3 inhibition (*Figure 2D*; see *Supplementary file 1* for quantitation of Western blots). Baseline c-MYC expression was also higher in hBMSC-CM compared to RPMI, and while complete suppression of c-MYC expression was observed at early time points after treatment with quizartinib in both conditions, c-MYC protein expression rebounded at later time points only in the presence of hBMSC-CM (*Figure 2D*; *Supplementary file 1*). We observed similar results in MV4-11 cells (*Figure 2—figure supplement 1*).

To explore the in vivo relevance of our findings, we performed a transcriptomic analysis of patient-derived primary FLT3-ITD$^+$ AML cells engrafted in NOD.*Rag1$^{-/-}$;γc$^{null}$* (NRG) mice expressing human cytokines GM-CSF, IL-3, and SCF (NRG-S mice). We chose to use NRG-S mice for our human xenograft mouse model because they not only express human cytokines previously shown to provide partial protection from FLT3 inhibition, but also demonstrate efficient engraftment of several patient-derived AML cells (*Barve et al., 2018*). To investigate the effect of the BM microenvironment on gene expression profiles of human AML cells, we compared transcriptomes of AML cells isolated from the spleen and the BM of mice treated with vehicle or quizartinib for 16 hr (*Figure 2E*). Cells isolated from the spleen showed a distinct gene expression signature compared to those from the BM, and treatment with quizartinib resulted in significant alterations of transcriptome profiles both in the spleen and the BM (*Figure 2F*). Our GSEA data showed multiple pathways with a similar gene expression pattern between AML cells from the spleen of quizartinib-treated mice and AML cells treated with quizartinib in RPMI, indicating similar effects of quizartinib on gene expression in both our in vitro and in vivo models (*Figure 2—figure supplement 2*). Next, we examined whether the BM microenvironment of NRG-S mice recapitulates the transcriptomic changes we observed in hBMSC-CM from our in vitro studies. Indeed, our GSEA data revealed that expression of mTORC1 signaling genes and MYC target genes was significantly more enriched in AML cells isolated from the BM than those from the spleen of quizartinib-treated mice, consistent with findings from our in vitro model (*Figure 2C and G*). Notably, genes of these pathways did not show significant differences between the BM and spleen of vehicle-treated mice, suggesting that higher expression of mTORC1 signaling and MYC target genes in the BM is specific to the context of FLT3 inhibition (*Figure 2—figure supplement 3*). In contrast, other pathways such as oxidative phosphorylation and G2/M checkpoint exhibited a similar enrichment in the BM compared to spleen for both vehicle and quizartinib-treated mice (*Figure 2G*, *Figure 2—source data 3*).

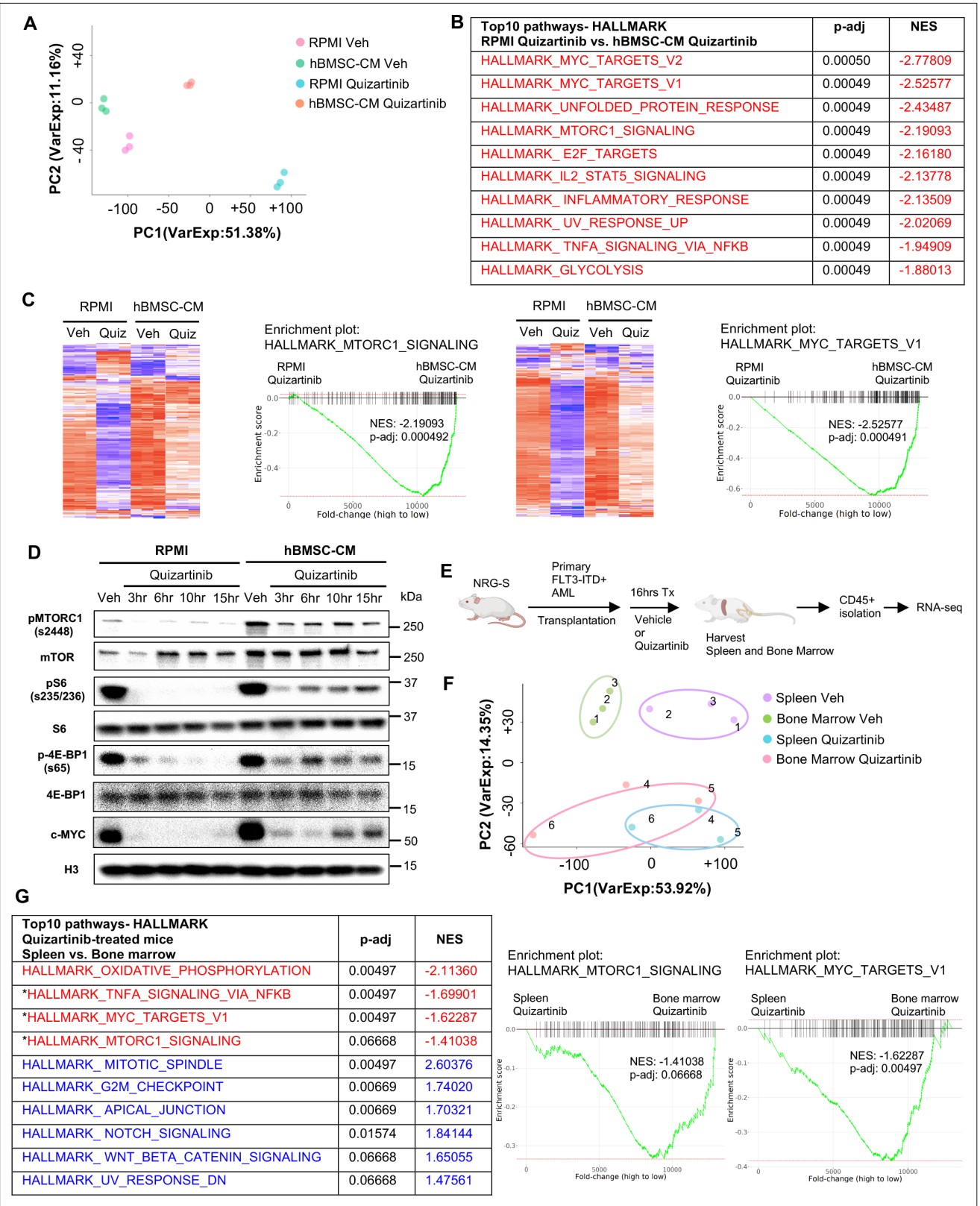

**Figure 2.** Bone marrow (BM) microenvironment substantially limits downregulation of mTOR and MYC pathway in FLT3-ITD acute myeloid leukemia (AML) cells upon FLT3 inhibition in vitro and in vivo. (**A–C**) RNA from MOLM-13 cells was harvested after 15 hr of treatment with vehicle or 3 nM quizartinib in either RPMI or human bone marrow stromal cell (hBMSC-CM) and subjected to RNA-seq (n = 3). (**A**) Principal component analysis (PCA) plot of each group. (**B**) Top 10 enriched signaling pathways (by p-adj) from gene set enrichment analyses (GSEA) of Hallmark gene sets applied on RNA-

*Figure 2 continued on next page*

*Figure 2 continued*

seq data from quizartinib-treated cells in hBMSC-CM compared with quizartinib-treated cells in RPMI. Pathways with significant alteration (p-adj<0.1) and negative normalized enrichment score (NES) are represented with red. (**C**) Enrichment plots and heatmaps showing comparisons of transcriptomes of mTORC1 signaling or MYC targets between quizartinib (Quiz)-treated cells in hBMSC-CM and quizartinib-treated cells in RPMI. (**D**) MOLM-13 cells were treated as indicated and harvested at different time points (3, 6, 10, and 15 hr) to measure mTORC1 signaling and c-MYC protein levels by Western blotting. (**E**) Design of in vivo study using human AML xenograft mouse model. NRG-S mice were transplanted with primary FLT3-ITD⁺ AML cells (AML #1: from a 69-year-old female patient with newly diagnosed AML with FLT3-ITD and NPM1 mutations). One cohort of mice was treated with vehicle, while the other cohort was treated with 2.5 mg/kg quizartinib for 16 hr (n = 3). After spleen and BM were harvested, human AML cells were isolated by CD45 isolation kit and RNA from each sample were collected for RNA-seq. (**F**) PCA plot of RNA-seq data from human AML samples isolated from spleen and BM. Samples from the same mouse are denoted by the same number. (**G**) Top 10 enriched signaling pathways (by p-adj) from GSEA of Hallmark gene sets applied on RNA-seq data from human AML cells. Samples from the BM of quizartinib-treated mice were compared with the samples from the spleen of the same paired mouse, and top 10 pathways are shown on the left. Pathways with significant alteration (p-adj<0.1) are represented with colors (red: negative NES; blue: positive NES). Asterisks denote pathways that exhibited similarly significant enrichment patterns from in vitro study (**B**). Enrichment plots of mTORC1 signaling or MYC target genes are shown on the right.

The online version of this article includes the following source data and figure supplement(s) for figure 2:

**Source data 1.** Unedited raw blots for *Figure 2D* are shown.

**Source data 2.** Unedited blots with labels for *Figure 2D* are shown.

**Source data 3.** Oxidative phosphorylation and G2/M checkpoint exhibited a higher enrichment in the bone marrow (BM) compared to spleen for both vehicle- and quizatinib-treated mice.

**Figure supplement 1.** Human bone marrow stromal cell (hBMSC-CM) substantially limits downregulation of mTOR and MYC pathway in MV4-11 cells upon FLT3 inhibition.

**Figure supplement 2.** Similar pathways are altered in FLT3-ITD acute myeloid leukemia (AML) cells following FLT3 inhibition in vitro and in vivo in the absence of bone marrow (BM) microenvironment.

**Figure supplement 2—source data 1.** Unedited raw blots for *Figure 2—figure supplement 2* are shown.

**Figure supplement 2—source data 2.** Unedited blots with labels for *Figure 2—figure supplement 2* are shown.

**Figure supplement 3.** Higher expression of mTORC1 signaling and MYC target genes in the bone marrow (BM) is specific to the context of FLT3 inhibition.

Collectively, our in vitro and in vivo models demonstrate consistent gene expression patterns, indicating that the BM microenvironment promotes maintenance of mTOR signaling and MYC target gene signatures in FLT3-ITD AML cells following FLT3 inhibition.

## Targeting the mTOR pathway reverses bone marrow-mediated protection of FLT3-ITD AML cells from FLT3 inhibition

As a downstream effector of activated FLT3 kinase, mTOR signaling has been implicated in the survival of FLT3-ITD⁺ AML cells, and aberrant activation of the mTOR pathway has been reported in FLT3-ITD AML cell lines that developed intrinsic resistance to FLT3 inhibitors through long-term culture with FLT3 inhibitor, coinciding with enhanced sensitivity to the combination of FLT3 and mTOR inhibitors (*Lindblad et al., 2016*; *Damnernsawad et al., 2022*). Our findings from unbiased studies using RPPA (*Figure 1E*) and RNA-seq (*Figure 2*) also suggest the potential role of the mTOR pathway in the BM-mediated survival of FLT3-ITD AML cells upon FLT3 inhibition. We first determined whether pharmacological inhibition of mTOR signaling reverses hBMSC-CM-mediated protection from apoptosis following FLT3 inhibition. Our data revealed that while the mTOR inhibitor everolimus alone did not significantly promote apoptosis, combinatorial treatment with everolimus and quizartinib significantly reversed hBMSC-CM-mediated protection from apoptosis mediated by quizartinib alone (*Figure 3A*). Furthermore, AML cells treated with quizartinib/everolimus in hBMSC-CM failed to recover after removal of drugs, in contrast to cells treated with quizartinib alone (*Figure 3B*). Similar results were observed in FLT3-ITD MV4-11 cells (*Figure 3—figure supplement 1*). In addition, combinatorial treatment of everolimus with a range of quizartinib concentrations demonstrated significantly greater elimination than quizartinib alone in both regular media RPMI and hBMSC-CM (*Figure 3—figure supplement 2*), although the extent of this protection by hBMSC-CM was reduced at the highest dose of quizartinib, potentially due to off-target effects. To assess whether normal human hematopoietic progenitors would be affected by the combination therapy, we treated CD34⁺ selected cord blood cells from a healthy newborn with quizartinib and/or everolimus, followed by plating on

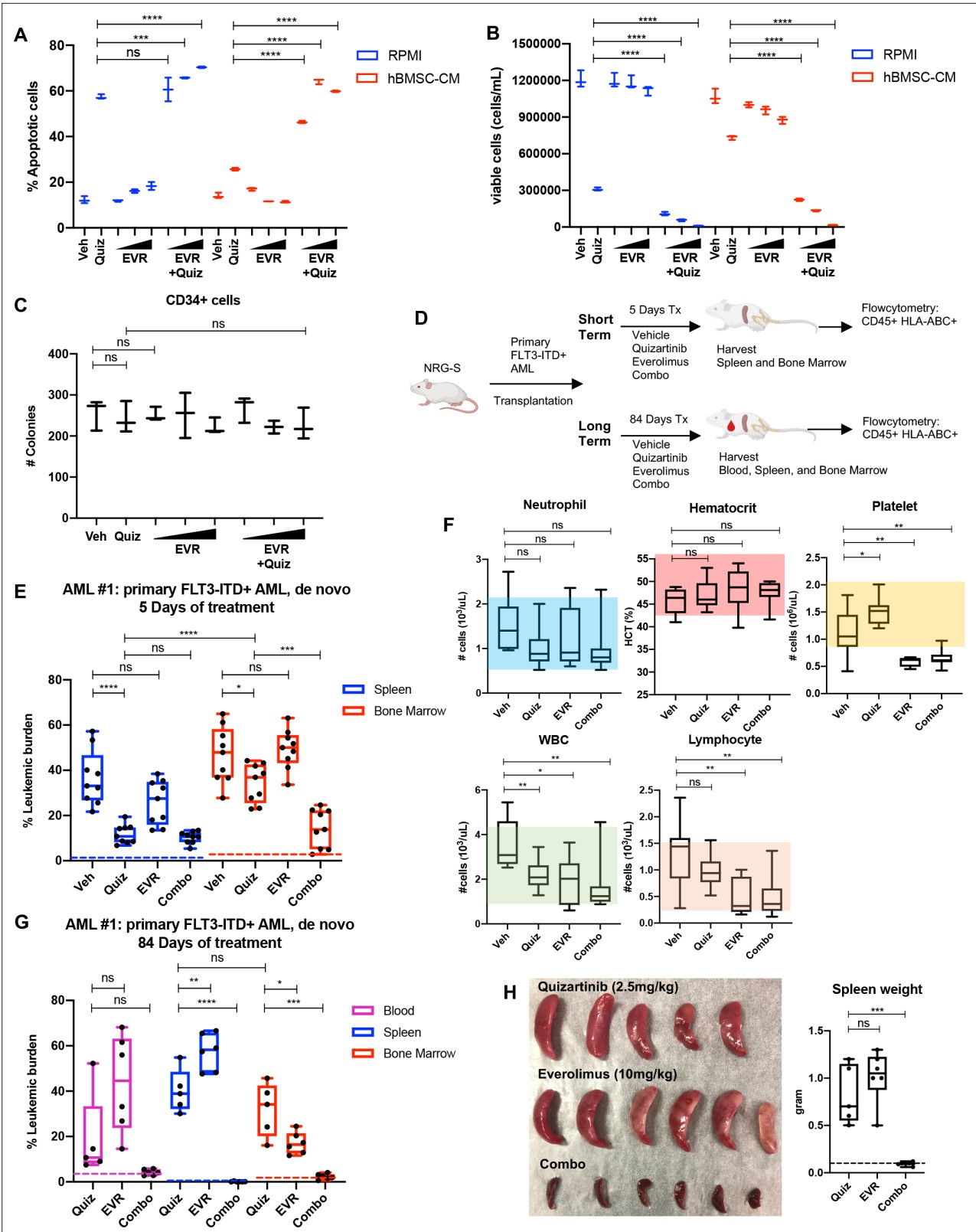

**Figure 3.** Targeting the mTOR pathway reverses the bone marrow (BM)-mediated protection of FLT3-ITD acute myeloid leukemia (AML) cells from FLT3 inhibition. (**A, B**) MOLM-13 cells were treated with 3 nM quizartinib (Quiz) and everolimus (EVR) alone or together as indicated (increasing concentrations of everolimus of 5, 10, and 50 nM indicated by triangles) for 48 hr in RPMI or human bone marrow stromal cell (hBMSC-CM) (n = 3). (**A**) Apoptosis was measured post-treatment, and (**B**) cell viability was measured after 5 days of rebound growth post-treatment. (**C**) CD34+ cord blood

*Figure 3 continued on next page*

*Figure 3 continued*

cells from a healthy newborn were treated with 10 nM quizartinib and everolimus alone or together as indicated (increasing concentrations of everolimus of 10, 50, and 100 nM indicated by triangles) for 48 hr. 25,000 cells from each group were seeded in methylcellulose post-drug treatments. After 11 days, the number of colonies was counted (n = 3). (**D**) Schematic diagram of experimental design. NRG-S mice were transplanted with human primary FLT3-ITD$^+$ AML (AML #1: a 69-year-old female patient who was newly diagnosed with AML with FLT3-ITD and NPM1 mutations). When leukemic burden reached 2–5% (~21–28 days after transplantation, determined by measuring CD45$^+$ HLA-ABC$^+$ cells in the blood), mice were divided into two groups. For short-term treatment, mice were treated with (1) vehicle, (2) 2.5 mg/kg quizartinib, (3) 10 mg/kg everolimus, and (4) combo for 5 days, daily p.o. (n = 9). For long-term treatment, mice were treated with the same regimen for 84 days, daily p.o. (n = 7–10). After treatment, leukemic burden was measured by quantifying HLA-ABC$^+$ CD45$^+$ cells via flow cytometry in spleen, BM, and blood (for long-term treatment only). (**E**) Plot showing leukemic burden of short-term treatment group; dotted line represents negative controls from non-transplanted mice. (**F**) Complete blood count (CBC) from blood samples of mice from long-term treatment group at day 5 of treatment is plotted. Colored boxes represent reference values (neutrophil: 0.54–2.15 × 10$^3$ cells/µL, hematocrit: 44.1–57.2%, platelet: 0.914–2.055 × 10$^3$ cells/µL, WBC: 0.94–4.12 × 10$^3$ cells/µL, lymphocyte: 0.26–1.56 × 10$^3$ cells/µL). (**G**) Plot showing leukemic burden of long-term treatment group, dotted line represents negative controls from nontransplanted mice. (**H**) Photograph of spleens from long-term treatment group on the day of harvest. Spleen weight from each group (dotted line represents average spleen weight from healthy untreated mice) is shown plotted on the right.

The online version of this article includes the following figure supplement(s) for figure 3:

**Figure supplement 1.** Combinatorial treatment of quizartinib with everolimus demonstrates significantly better cell-killing effects in MV4-11 cells.

**Figure supplement 2.** Combinatorial treatment of everolimus with a range of quizartinib concentrations demonstrated significantly greater elimination than quizartinib alone in both regular media RPMI and human bone marrow stromal cell (hBMSC-CM).

**Figure supplement 3.** Combinatorial treatment of quizartinib with everolimus does not exhibit reductions in weight.

**Figure supplement 4.** Combinatorial treatment of quizartinib with everolimus demonstrates significantly better cell-killing effects in xenograft mouse model utilizing the primary acute myeloid leukemia (AML) from a relapse patient.

methylcellulose for clonogenic assays. None of these treatments showed significant changes in colony numbers, suggesting minimal effects on healthy hematopoietic progenitors (*Figure 3C*).

To test whether inhibition of mTOR could enhance the efficacy of quizartinib in eliminating FLT3-ITD$^+$ AML cells in vivo, we tested the combination therapy in our xenograft mouse model. As illustrated in *Figure 3D*, mice engrafted with patient-derived FLT3-ITD$^+$ AML cells were treated with vehicle, quizartinib, or everolimus alone, or these drugs in combination for 5 days to assess the short-term effects of these therapies on leukemic burden in the spleen and BM. A separate cohort of mice were treated with the same drug regimen for 84 days to examine the long-term effects of the therapies. After 5 days of treatment, quizartinib alone substantially reduced AML cells in the spleen, but only modestly reduced AML burden in the BM (*Figure 3E*). On the other hand, the addition of everolimus to quizartinib ('combo' therapy) resulted in effective elimination of leukemia burden in the BM, recapitulating our in vitro data with hBMSC-CM. Complete blood counts revealed that mice treated with everolimus alone or in combination with quizartinib showed a modest level of thrombocytopenia and leukopenia without affecting neutrophils or hematocrit (*Figure 3F*), consistent with patient data that everolimus has been reported to induce hematological changes that are tolerable (*Chen et al., 2015a*). Mice on the long-term treatments did not exhibit reductions in weight (*Figure 3—figure supplement 3*). Throughout the duration of long-term treatment, vehicle-treated mice had a median survival of 27 days and all had succumbed by 84 days. While mice receiving long-term treatment with quizartinib alone eventually relapsed with high leukemic burden detected in the blood, BM, and spleen, leukemia cells were not detected in any compartment from mice treated with the combination therapy (*Figure 3G*). Elimination of the leukemia in the spleen by the combination therapy was further confirmed by the absence of splenomegaly (*Figure 3H*). Furthermore, in an independent experiment with a second primary FLT3-ITD+ AML sample from a patient who developed relapse after chemotherapy, we observed potent elimination of leukemic burden in all compartments with the combination therapy. While monotherapy with quizartinib was effective in peripheral tissues (spleen and blood), effective elimination of leukemia cells in the BM was only achieved with the combinatorial treatment of quizartinib with everolimus. Notably, everolimus alone demonstrated similar effectiveness as quizartinib monotherapy in eliminating leukemia cells only in the blood without showing significant effect in the spleen or BM, suggesting that mTOR inhibition alone results in different cell-killing effects depending upon tissue microenvironment (*Figure 3—figure supplement 4*). As before, we did not observe significant alterations in mouse weights, indicating that the therapy was well tolerated (*Figure 3—figure supplement 4*).

In all, these data show that mTOR pathway plays a key role in the BM-mediated protection of FLT3-ITD AML cells from FLT3-targeted therapy.

## Suppression of protein translation upon FLT3 inhibition is significantly alleviated by hBMSC-CM via an mTOR-dependent mechanism

Control of protein translation is known to be one of the critical proliferation pathways mediated by mTOR signaling in cancer cells (*Saxton and Sabatini, 2017*). In particular, regulation of protein synthesis via mTOR signaling has been implicated in human myeloid leukemogenesis, making it an attractive potential therapeutic target in AML cells (*Tamburini et al., 2009*). Therefore, we speculated that altered protein translation would be an important downstream effect of mTOR signaling that drives the BM-mediated survival of FLT3-ITD+ AML cells following FLT3 inhibition. Indeed, our transcriptome analysis revealed significantly higher expression of genes involved in translation regulation in cells treated with quizartinib in hBMSC-CM than those in RPMI (*Figure 4A*). Moreover, FLT3 inhibition in RPMI dramatically reduced phosphorylation of 4E-BP1 and ribosomal protein S6 that regulate translation downstream of mTORC1 signaling, while addition of hBMSC-CM led to partial maintenance of these activities (*Figure 2D* and *Figure 4B*), consistent with the partial restoration of mTOR activity. Importantly, dual inhibition of mTOR and FLT3 eliminates the effect of hBMSC-CM on maintenance of these translation regulators.

To directly assess the effects on protein translation, we measured translation activity in FLT3-ITD AML cells after each drug treatment by detecting cells labeled with O-propargyl-puromycin (OPP), a puromycin analog that is incorporated into newly translated proteins, allowing a reliable quantification of protein translation (*Nagelreiter et al., 2018*). Strikingly, FLT3 inhibition alone was sufficient to dramatically reduce translation activity in FLT3-ITD AML cells over the course of treatment, comparable to the effect of translation inhibition by cycloheximide, but the reduction was significantly limited by hBMSC-CM. Moreover, combined inhibition of mTOR and FLT3 reversed the partial restoration of translation activity in hBMSC-CM, indicating that the effect of hBMSC-CM on translation activity is dependent on restored mTOR activity (*Figure 4C*). The number of viable cells was maintained in every treatment condition through this time course, indicating that the significant reduction in translation activity was not due to concurrent cell death (*Figure 4—figure supplement 1*). Similarly, MV4-11 cells also showed mTOR-dependent partial restoration of translation activity in hBMSC-CM. The degree of restoration was less than what we observed from MOLM-13 cells, consistent with the lesser extent of hBMSC-CM-mediated protection from apoptosis following FLT3 inhibition (*Figure 4—figure supplement 2*).

Following FLT3 inhibition, a substantial population of AML cells showed a very low level of OPP labeling (OPPlow) in RPMI, indicating severe suppression of translation, while hBMSC-CM significantly reduced the proportion of OPPlow cells and the OPPhigh cells showed a noticeable leftward peak shift compared to vehicle-treated cells (consistent with a partial inhibition of translation on a per-cell basis). Furthermore, dual inhibition of mTOR and FLT3 by combination treatment resulted in a dramatic increase of OPPlow cells even in the presence of hBMSC-CM (*Figure 4D*). Our data from earlier time points suggest that leftward peak shift in OPPhigh cells and accumulation of OPPlow cells occur over the course of drug treatment (*Figure 4—figure supplement 3*). Notably, mTOR inhibition alone showed a leftward shift in OPPhigh cells without noticeable detection of OPPlow cells (*Figure 4—figure supplement 3*), consistent with a partial reduction in overall translation (that apparently is well-tolerated by cells).

To determine how hBMSC-CM affects translation of specific mRNAs in FLT3-ITD AML cells following inhibition of FLT3 and/or mTOR signaling, we performed polysome fractionization using ultracentrifugation through sucrose density gradients followed by analysis of mRNA distribution profiles. As expected, cells treated with quizartinib in RPMI showed a much higher proportion of transcripts bound to the 80S monosome and substantially reduced polysome-associated transcripts, consistent with impaired translation activity (*Figure 4E*). In contrast, FLT3 inhibition in hBMSC-CM showed partial restoration of polysome content, especially heavy polysomes that consist of five or more ribosomes, with a concomitant decrease in the 80S monosome peak. Moreover, addition of everolimus to quizartinib treatment reversed the partial restoration of heavy polysomes in hBMSC-CM. Findings from our polysome profiling are consistent with what we observed from OPP assays, further validating the role of restored mTOR signaling by hBMSC-CM in regulation of protein translation. Finally, we

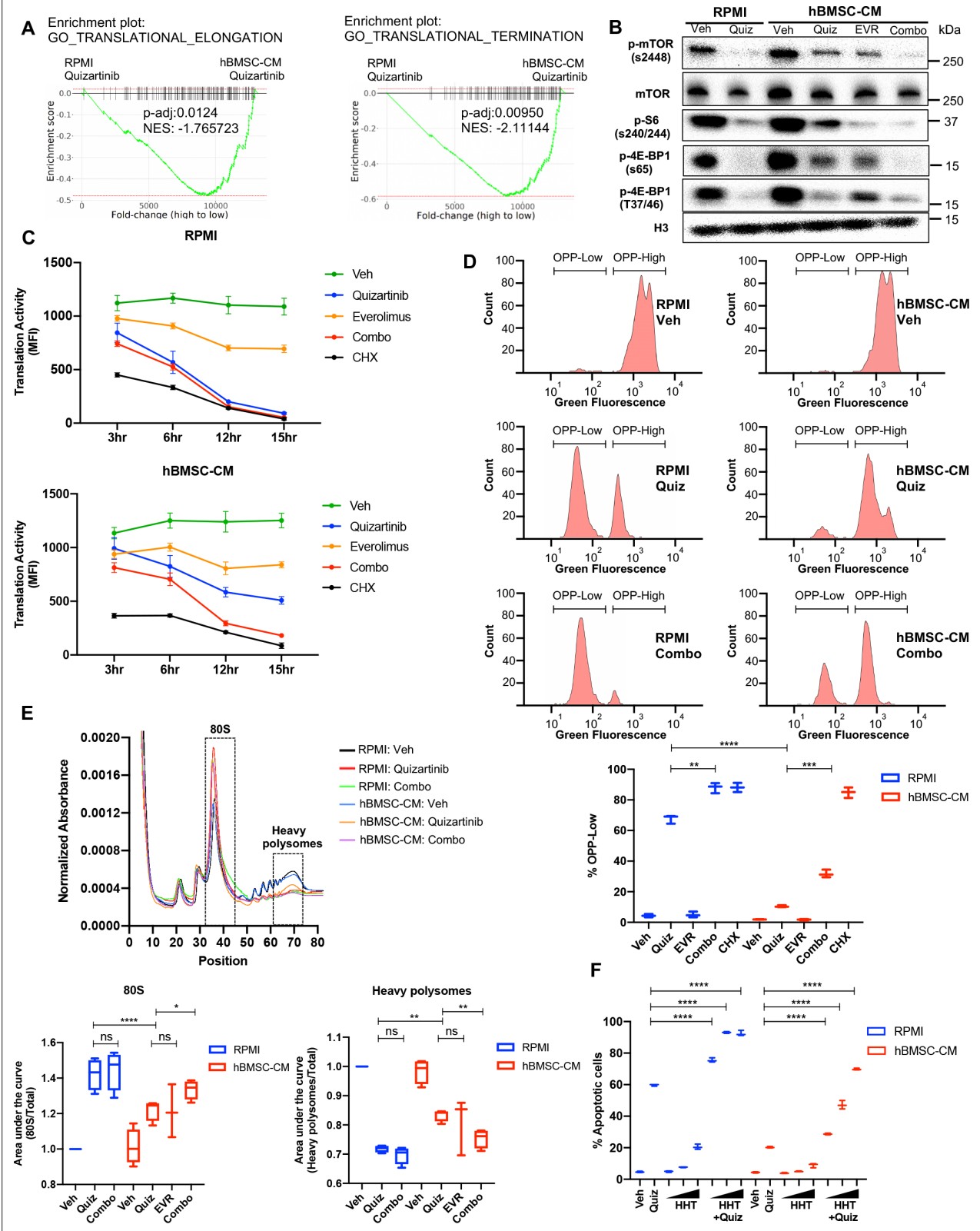

**Figure 4.** Suppression of protein translation upon FLT3 inhibition is significantly alleviated by human bone marrow stromal cell (hBMSC-CM) via an mTOR-dependent mechanism. (**A**) Gene set enrichment analyses (GSEA) of Gene Ontology gene sets from in vitro study (*Figure 2A–C*) showing enrichment of translational control in acute myeloid leukemia (AML) cells treated with quizartinib in hBMSC-CM, compared with quizartinib-treated cells in RPMI. (**B**) MOLM-13 cells were treated with vehicle, 3 nM quizartinib, 10 nM everolimus, or quizartinib + everolimus (combo) in RPMI or

*Figure 4 continued on next page*

*Figure 4 continued*

hBMSC-CM for 15 hr and harvested for Western blotting to measure mTOR signaling and its downstream mediators involved in translational control. (**C**) MOLM-13 cells were treated with the indicated drugs at different time points (3, 6, 12, and 15 hr), followed by labeling with O-propargyl-puromycin (OPP), to measure translation activity using flow cytometry (MFI: mean fluorescence intensity for OPP incorporation). Cells were treated with 10 ug/mL cycloheximide (CHX) as a control for inhibition of translation (n = 3). (**D**) Representative flow cytometry histograms of OPP assay from cells treated with vehicle, quizartinib, and combo in RPMI or hBMSC-CM for 15 hr, showing binary peaks (OPP-High and OPP-Low). Percent of OPP-Low population from all treatment groups is shown at the bottom. (**E**) MOLM-13 cells were treated with the indicated drugs for 12 hr and harvested for polysome profiling. Area under the curve for each treatment was normalized to the vehicle-treated group in RPMI and is shown for 80S peak fraction (position 33–44) and heavy polysome fraction (position 62–73) at the bottom (n = 3 for everolimus-treated cells in hBMSC-CM, n = 4 for all other groups). (**F**) MOLM-13 cells were treated with 3 nM quizartinib and homoharringtonine (HHT) alone or together as indicated (increasing concentrations of HHT of 1, 5, and 10 nM indicated by triangles) for 48 hr in RPMI or hBMSC-CM, followed by measurement of apoptosis (n = 3).

The online version of this article includes the following source data and figure supplement(s) for figure 4:

**Source data 1.** Unedited raw blots for *Figure 4B* are shown.

**Source data 2.** Unedited blots with labels for *Figure 4B* are shown.

**Figure supplement 1.** FLT3-ITD acute myeloid leukemia (AML) cells demonstrate shift of O-propargyl-puromycin (OPP)-labeled peaks over the course of drug treatments without showing signs of cell death.

**Figure supplement 2.** Suppression of protein translation upon FLT3 inhibition is significantly alleviated by human bone marrow stromal cell (hBMSC-CM) via an mTOR-dependent mechanism in MV4-11 cells.

**Figure supplement 3.** Leftward peak shift in OPP^high cells and accumulation of OPP^low cells occur over the course of drug treatment.

tested whether pharmacological inhibition of translation recapitulates the effect of mTOR inhibition in reversing hBMSC-CM-mediated protection from apoptosis following FLT3 inhibition. Indeed, combinatorial treatment of quizartinib with the translation inhibitor homoharringtonine (HHT) showed a significantly higher number of apoptotic cells compared to cells treated with quizartinib alone, even in the presence of hBMSC-CM (*Figure 4F*), phenocopying the effect of everolimus. These results further support previously reported synergistic elimination of FLT3-ITD AML cells following treatment with FLT3 inhibitors in combination with HHT (*Lam et al., 2016*). Taken together, our data suggest that mTOR-dependent partial restoration of translation activity is a key survival mechanism of FLT3-ITD AML cells in hBMSC-CM following FLT3 inhibition.

## Examination of polysome enrichment of transcripts reveals a critical role for oxidative phosphorylation in hBMSC-CM-mediated protection from FLT3 inhibition

Our OPP assays and polysome profiling data indicate that hBMSC-CM induces partial restoration of translation activity and polysome-associated transcripts following FLT3 inhibition, respectively. Yet, the mechanism for how cells survive despite incomplete restoration of translation remains unknown. To investigate this, we further analyzed mRNAs enriched in heavy polysomes to examine whether specific subsets of mRNAs were selectively translated. First, we measured levels of RNAs isolated from the whole lysates (input) and heavy polysome fractions. As expected, cells treated with quizartinib alone or quizartinib and everolimus in combination demonstrated a significant reduction of RNA levels that was substantially more extensive in polysomes than inputs (*Figure 5A*), consistent with the suppressive effects of FLT3 inhibition on polysome engagement of transcripts. mRNAs isolated from input and heavy polysomes were subjected to RNA-seq. Notably, GSEA applied on RNA-seq data from input (unfractionated) samples indicated that the addition of everolimus to quizartinib in hBMSC-CM significantly suppressed a majority of the pathways that were shown to be restored by hBMSC-CM following treatment of quizartinib alone, including genes for mTORC1 signaling, MYC targets, oxidative phosphorylation, and translation regulation (*Figure 5B and C*; compare to *Figure 2B and G*). Thus, mTOR signaling is essential for the ability of hBMSC-CM to maintain the expression of these key pathways in FLT3-inhibited cells.

In order to identify transcripts that are preferentially retained or depleted from polysomes, we performed GSEA on mRNAs in heavy polysomes normalized to input. Remarkably, only one Hallmark pathway, oxidative phosphorylation, exhibited significant enrichment in heavy polysomes isolated from cells treated with quizartinib in hBMSC-CM compared to those in RPMI (*Figure 5D*). Furthermore, combinatorial treatment of quizartinib with everolimus significantly reduced polysome enrichment

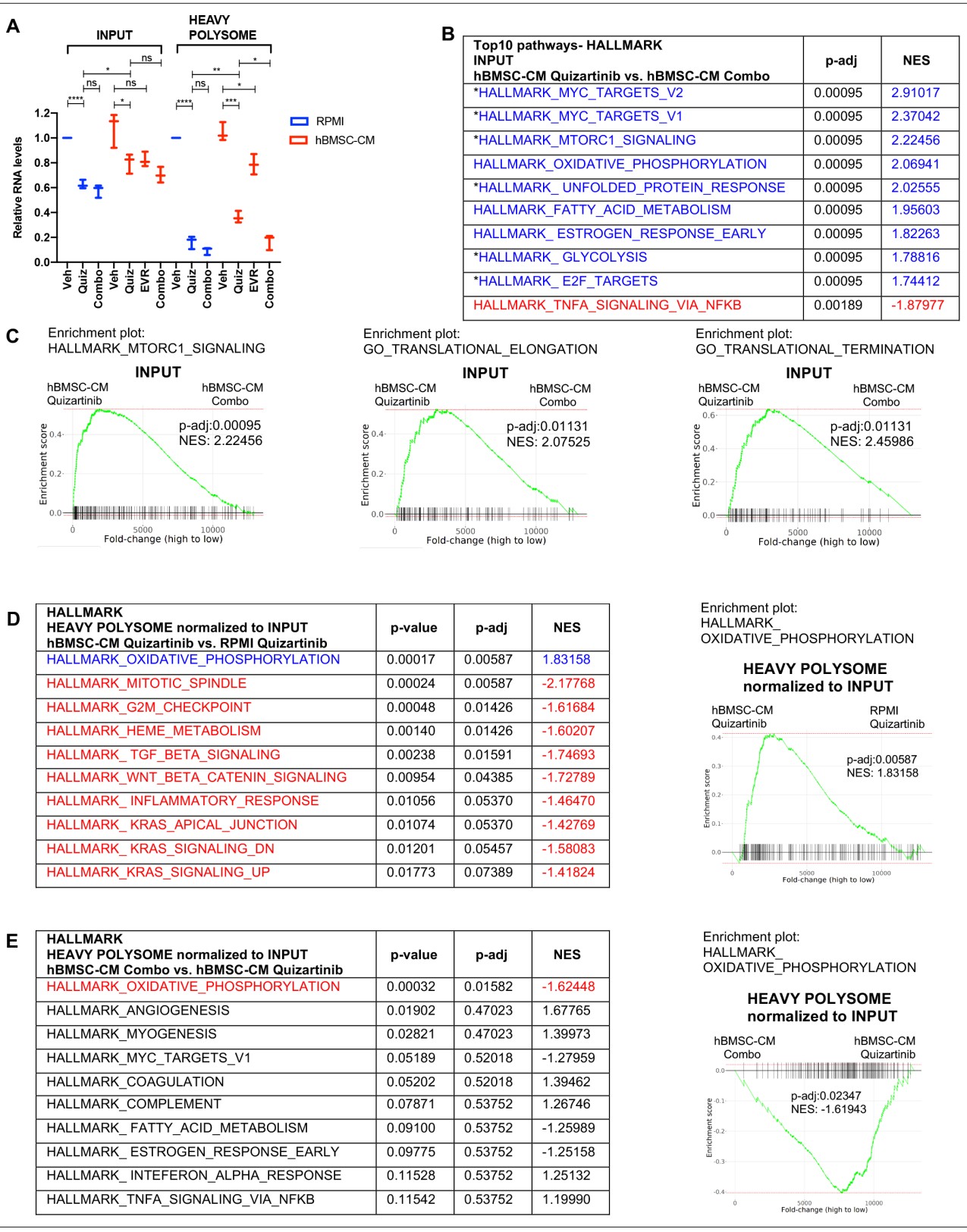

**Figure 5.** Polysome enrichment of transcripts by human bone marrow stromal cell (hBMSC-CM) following FLT3 inhibition is highly selective for oxidative phosphorylation mRNAs and is mTOR-dependent. (**A**) RNA yields from whole-cell lysates (input) and heavy polysome fractions (*Figure 4E*) were measured and normalized to vehicle-treated group in RPMI for each biological replicate. (**B, C**) Gene set enrichment analyses (GSEA) of Hallmark and Gene Ontology gene sets was performed on RNA-seq data from input samples. Quizartinib-treated cells in hBMSC-CM were compared with combo-

*Figure 5 continued on next page*

*Figure 5 continued*

treated cells in hBMSC-CM. (**B**) Top 10 enriched signaling pathways from GSEA of Hallmark gene sets with significant alterations (p-adj<0.1) represented with colors (red: negative normalized enrichment score [NES]; blue: positive NES). Asterisks denote pathways that exhibited partial restoration following quizartinib treatment in hBMSC-CM compared to those in RPMI (*Figure 2B*). (**C**) Enrichment plots of mTOR signaling, translational elongation, and translational termination are shown. (**D, E**) Top 10 enriched signaling pathways (by p-adj) from GSEA of Hallmark gene sets for RNA-seq data from human acute myeloid leukemia (AML) cells from heavy polysome-associated mRNAs normalized to input. Pathways with significant alteration (p-adj<0.1) are represented with colors (red: negative NES; blue: positive NES). Analyses were done by comparing (**D**) cells treated with quizartinib in hBMSC-CM with those in RPMI or (**E**) cells treated with combo in hBMSC-CM with those treated with quizartinib alone in hBMSC-CM. Enrichment plots of oxidative phosphorylation pathway for each comparison are shown on the right.

The online version of this article includes the following figure supplement(s) for figure 5:

**Figure supplement 1.** Transcripts associated with mitochondrial integrity are enriched in polysomes of FLT3-ITD acute myeloid leukemia (AML) cells in human bone marrow stromal cell (hBMSC-CM) upon FLT3 inhibition.

of oxidative phosphorylation mRNAs, even in the presence of hBMSC-CM, compared to quizartinib alone (*Figure 5E*). In fact, oxidative phosphorylation was the only Hallmark gene set with significant (by p-adj) alteration with the combination treatment. Interestingly, as shown in *Figure 2G*, oxidative phosphorylation was also the most enriched pathway identified in primary AML cells in the BM relative to the spleen in mice treated with quizartinib. Our GSEA of Hallmark gene sets demonstrates that other pathways with statistical significance showed opposite enrichment patterns, suggesting polysome enrichment of transcripts by hBMSC-CM following FLT3 inhibition is highly selective to oxidative phosphorylation mRNAs and is mTOR-dependent (*Figure 5D and E*). In accordance with this, GSEA of Gene Ontology gene sets demonstrated similar polysome enrichment patterns for transcripts that are involved with mitochondria integrity and biogenesis (*Figure 5—figure supplement 1*). Given that mTOR has been shown to control transcription of mitochondrial genes through a complex involving transcription factors such as PGC1-α and YY1 (*Cunningham et al., 2007*), we further compared levels of transcripts in input samples of multiple mitochondrial genes and transcription factors that regulate the expression of mitochondrial genes. Interestingly, PGC1-α and the mitochondrial enzyme IDH3A demonstrated mTOR-dependent restoration by hBMSC-CM following FLT3 inhibition (*Figure 5—figure supplement 1*). PGC1-α is a transcriptional coactivator, and thus its mTOR-dependent restoration by hBMSC-CM could contribute to the partial restoration of OXPHOS, a possibility that would require further investigation. Notably, other mitochondrial transcriptional regulators and mitochondrial genes examined did not show such patterns.

To examine whether altered expression and polysome enrichment of oxidative phosphorylation mRNAs translates into corresponding changes in mitochondrial respiration, we measured the oxygen consumption rate (OCR) of MOLM-13 cells following inhibition of FLT3 and/or mTOR in either RPMI or hBMSC-CM. In agreement with our findings from GSEA, we found a dramatic decrease in OCR in cells following FLT3 inhibition, while the OCR reduction was partially restored in the presence of hBMSC-CM (*Figure 6A*). Furthermore, the dual inhibition of FLT3 and mTOR reversed this partial restoration of OCR. Of note, the degree of restoration was greater for maximal respiration than basal respiration, as represented by reserve respiratory capacity that reflects the difference between maximal and basal respiratory rate (*Figure 6A*). The observed changes in OCR coincide with similar alterations in ATP levels (*Figure 6B*), indicating that restoration of OCR is able to rescue ATP, which may be key for cell survival. Similarly, we observed hBMSC-CM-mediated partial restoration of OCR following FLT3 inhibition in MV4-11 cells. While the degree of restoration is less than what we observed in MOLM-13 cells, the dual inhibition of FLT3 and mTOR reversed this restoration of OCR. Moreover, alterations in OCR translate into corresponding changes in ATP levels in MV4-11 cells as well (*Figure 6—figure supplement 1*). Indeed, combinatorial treatment of quizartinib with ATP synthase inhibitor oligomycin A significantly reversed hBMSC-CM-mediated protection from apoptosis (*Figure 6C*), consistent with the effect of everolimus or HHT (*Figures 3A and 4F*), as well as our previous study that showed that tyrosine kinase inhibition engenders acute sensitivity to oligomycin A in myeloid leukemias (*Alvarez-Calderon et al., 2015*).

Taken together, our data suggest that the polysome enrichment of oxidative phosphorylation mRNAs mediates survival of FLT3-ITD AML cells in hBMSC-CM following FLT3 inhibition through an mTOR-dependent mechanism, potentially by maintenance of energy levels.

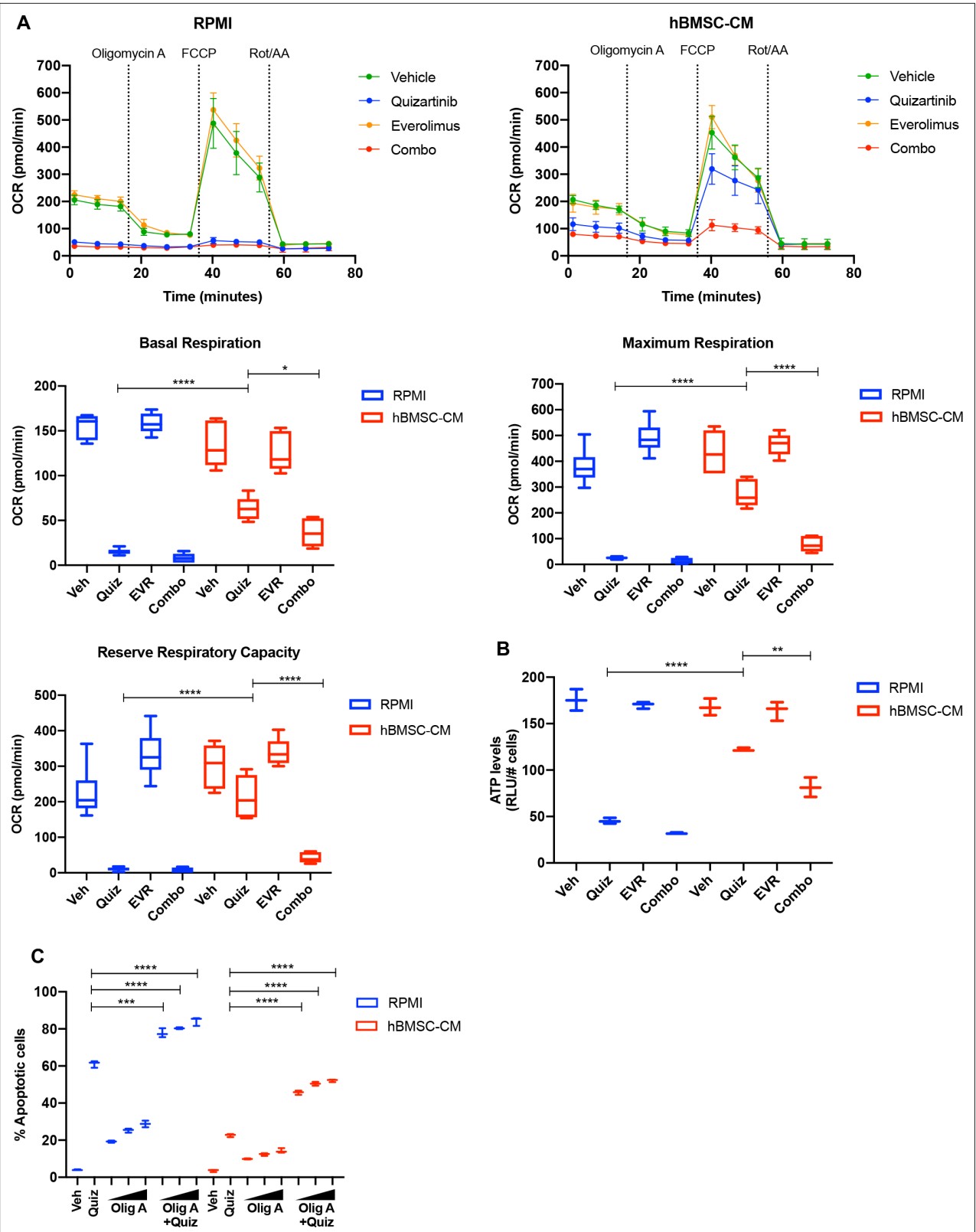

**Figure 6.** mTOR-dependent oxidative phosphorylation is a key survival mechanism of human bone marrow stromal cell (hBMSC-CM)-mediated protection from FLT3 inhibition. (**A**) MOLM-13 cells were treated with indicated drugs for 15 hr, followed by measurement of oxygen consumption rate (OCR) by Seahorse XF Cell Mito Stress Test (n = 5). Basal respiration, maximum respiration, and reserve respiratory capacity are shown at the bottom. (**B**) MOLM-13 cells were treated with indicated drugs for 15 hr, followed by luminescence-based ATP assays (normalizing relative light units [RLU] to cell

*Figure 6 continued*

number). (**C**) MOLM-13 cells were treated with 3 nM quizartinib and oligomycin A (Olig A) alone or together as indicated (increasing concentrations of Olig A of 1, 2, and 4 nM indicated by triangles) for 48 hr in RPMI or hBMSC-CM, followed by measurement of apoptosis (n = 3).

The online version of this article includes the following figure supplement(s) for figure 6:

**Figure supplement 1.** Human bone marrow stromal cell (hBMSC-CM)-mediated partial restoration of oxidative phosphorylation is mTOR-dependent in MV4-11 cells.

## ATM mediates the protection by hBMSC-CM upstream of mTOR signaling and oxidative phosphorylation

Having established as axis between mTOR activity and oxidative phosphorylation, we next sought to better understand factors upstream of mTOR that may regulate activity critical to survival of AML cells. Our previous work has shown that inhibition of ATM (ataxia telangiectasia mutated) in FLT3-ITD AML cells causes synergistic cell killing with FLT3 inhibitors by inducing apoptosis through exacerbation of mitochondrial oxidative stress (*Gregory et al., 2016*). Additionally, accumulating evidence indicates that ATM drives the BM-mediated resistance to chemotherapy in other hematological malignancies, such as multiple myeloma and acute lymphoblastic leukemia via upregulation of cytokines such as IL-6 (*Tang et al., 2018*) or CCL3, CCL4, and GDF15 (*Chen et al., 2019*). We found that blocking FLT3 inhibited ATM expression in RPMI, but that ATM expression was maintained in hBMSC-CM following FLT3 inhibition in FLT3-ITD AML cells (*Figure 7A*). However, dual inhibition of mTOR and FLT3 did not reverse the effect of hBMSC-CM on ATM levels, suggesting that ATM acts upstream of mTOR. We observed that phosphorylated ATM (S1981) was predominantly detected at a smaller size (~250 kDa) than full-length ATM (350 kDa) following FLT3 inhibition in RPMI, while hBMSC-CM significantly restored full-length phosphorylated ATM (*Figure 7A*; *Supplementary file 1*). Moreover, dual inhibition of mTOR and FLT3 resulted in a higher ratio of small to full-length phospho-ATM. Treatment of cells with pan-caspase inhibitor Z-VAD-FMK resulted in the elimination of the small ATM isoform following FLT3 inhibition, suggesting that cleavage of full-length ATM was mediated by caspase activity (*Figure 7—figure supplement 1*), as previously reported from other studies (*Smith et al., 1999*; *Wang et al., 2006*). Co-treatment of Z-VAD-FMK with quizartinib in RPMI partially rescued cell death (*Figure 7—figure supplement 1*).

Next, we examined the potential role of ATM in hBMSC-CM-mediated protection of FLT3-ITD AML cells following FLT3 inhibition. Compared to cells expressing nontargeting shRNA (shNT), cells with shRNA-mediated ATM knockdown (sh*ATM*) demonstrated reversal of protection against apoptosis upon FLT3 inhibition in hBMSC-CM (*Figure 7B*), as well as impaired recovery of cell growth after removal of quizartinib (*Figure 7—figure supplement 2*). We also observed reversal of hBMSC-CM-mediated protection with ATM knockdown in MV4-11 cells (*Figure 7—figure supplement 3*). Given that ATM has long been known to regulate cell cycle checkpoints, we asked whether knocking down ATM allowed cell cycle progression despite FLT3 inhibition via bypassing ATM-dependent checkpoint arrest. We found that sh*ATM* did not prevent the G1 arrest observed with quizartinib with or without hBMSC-CM (*Figure 7—figure supplement 4*), indicating that the mechanism of hBMSC-CM-mediated protection by ATM is not through activating cell cycle checkpoints.

Based on the importance of mTOR signaling in hBMSC-CM-mediated protection, we wanted to further investigate the potential association between ATM and mTOR signaling in FLT3-ITD AML cells. Strikingly, knockdown of ATM resulted in the complete absence of detectable mTOR protein expression (validated with two different antibodies as described in 'Materials and methods'), without significantly affecting the baseline phosphorylation of 4E-BP1 and S6 (*Figure 7C*). Still, upon FLT3 inhibition, sh*ATM* prevented the restoration of p-4E-BP1 or p-S6 otherwise elicited by hBMSC-CM. Additionally, sh*ATM* not only resulted in a moderate reduction of AKT protein levels, but also impaired the hBMSC-CM-mediated restoration of phosphorylation of AKT following FLT3 inhibition (*Figure 7C*; *Supplementary file 1*). These results implicate ATM as a key regulator of mTORC1 signaling through regulation of the mTOR protein itself, as well as AKT that has been shown to regulate mTORC1 via phosphorylation (*Saxton and Sabatini, 2017*). It is important to note that we observed the same results in CRISPR/Cas9-mediated ATM knockout FLT3-ITD AML cells (*Figure 7—figure supplement 5*). The ATM knockout cells also showed greater sensitivity to FLT3 inhibition in RPMI and reversal of protection against apoptosis upon FLT3 inhibition in hBMSC-CM, compared to control cells (*Figure 7—figure*

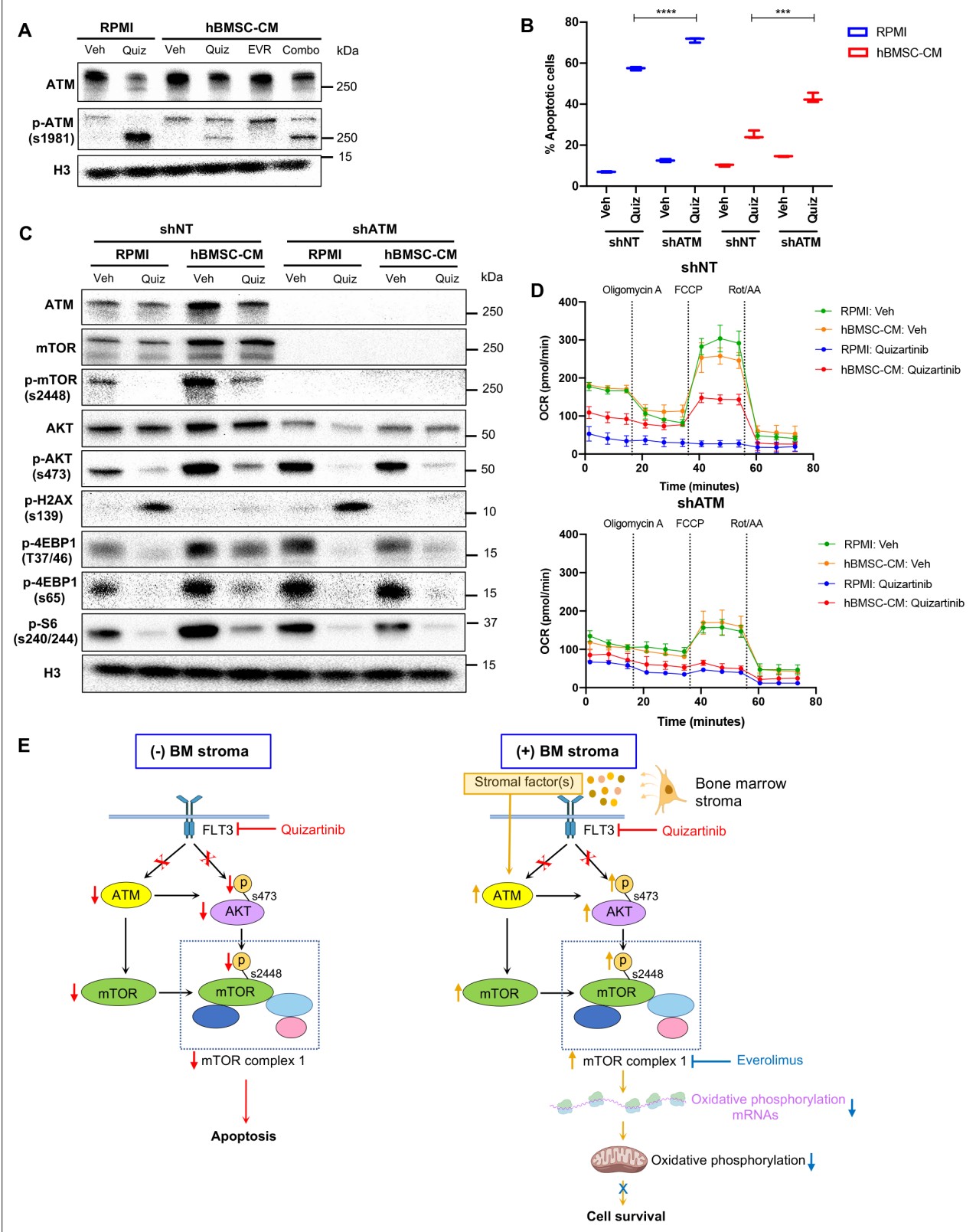

**Figure 7.** ATM mediates the protection from FLT3 inhibition by human bone marrow stromal cell (hBMSC-CM) upstream of mTOR signaling and oxidative phosphorylation. (**A**) MOLM-13 cells were treated with vehicle, 3 nM quizartinib, 10 nM everolimus, and combo in RPMI or hBMSC-CM for 15 hr and harvested to determine the levels of total and phosphorylated ATM by Western blotting. (**B**) MOLM-13 cells were transduced with lentiviruses-expressing shRNAs (shNT: nontargeting shRNA; or sh*ATM*: ATM-targeting shRNA). Cells were treated with either vehicle or 3 nM quizartinib (Quiz)

*Figure 7 continued on next page*

*Figure 7 continued*

for 48 hr in RPMI or hBMSC-CM, followed by measurement of apoptosis (n = 3). (**C**) MOLM-13 cells expressing shNT or sh*ATM* were treated with the indicated conditions for 15 hr and harvested for Western blot analyses. (**D**) MOLM-13 cells expressing shNT or sh*ATM* were treated with the indicated conditions for 15 hr, followed by measurement of oxygen consumption rate (OCR) by Seahorse XF Cell Mito Stress Test (n = 4). (**E**) Model of ATM and mTOR-dependent survival of FLT3-ITD acute myeloid leukemia (AML) cells following FLT3 inhibition in BM microenvironment. In the absence of BM stroma, FLT3-ITD AML cells undergo apoptosis upon FLT3 inhibition as a result of downregulation of ATM and mTOR complex 1 (mTORC1) activity (left). In the presence of BM stroma, on the other hand, FLT3-ITD AML cells are protected from cell-killing effects by FLT3 inhibition (right). As indicated by orange arrows, the BM stroma-mediated protection is driven by ATM through regulation of mTOR protein levels, as well as mTORC1 activity via AKT, subsequently leading to partial restoration of translation. Restored mTORC1 activity selectively drives translation of oxidative phosphorylation mRNAs that promote cell survival even in the presence of FLT3 inhibition. The BM stroma-mediated protection against apoptosis upon FLT3 inhibition is reversed with combinatorial treatment with everolimus, which inhibits mTOR signaling and downstream processes including polysome enrichment and selective translation of oxidative phosphorylation mRNAs, and oxidative phosphorylation. Effects of drugs that target FLT3 (quizartinib) or mTOR (everolimus) are represented with red or blue, respectively, arrows and Xs.

The online version of this article includes the following source data and figure supplement(s) for figure 7:

**Source data 1.** Unedited raw blots for *Figure 7A* are shown.

**Source data 2.** Unedited blots with labels for *Figure 7A* are shown.

**Source data 3.** Unedited raw blots for *Figure 7C* are shown.

**Source data 4.** Unedited blots with labels for *Figure 7C* are shown.

**Figure supplement 1.** Cleavage of full-length ATM is mediated by caspase activity upon FLT3 inhibition.

**Figure supplement 1—source data 1.** Unedited raw blots for *Figure 7—figure supplement 1* are shown.

**Figure supplement 1—source data 2.** Unedited blots with labels for *Figure 7—figure supplement 1* are shown.

**Figure supplement 2.** sh*ATM* further impairs recovery of cell growth after removal of quizartinib.

**Figure supplement 3.** ATM mediates the protection from FLT3 inhibition by human bone marrow stromal cell (hBMSC-CM) in MV4-11 cells.

**Figure supplement 4.** sh*ATM* does not prevent the G1 arrest upon FLT3 inhibition.

**Figure supplement 5.** ATM knockout by CRISPR/Cas9 phenocopies sh*ATM*.

**Figure supplement 5—source data 1.** Unedited raw blots for *Figure 7—figure supplement 5* are shown.

**Figure supplement 5—source data 2.** Unedited blots with labels for *Figure 7—figure supplement 5* are shown.

**Figure supplement 6.** ATM-dependent mTOR expression and AKT phosphorylation are not unique to FLT3-dependent acute myeloid leukemia (AML).

**Figure supplement 6—source data 1.** Unedited raw blots for *Figure 7—figure supplement 6* are shown.

**Figure supplement 6—source data 2.** Unedited blots with labels for *Figure 7—figure supplement 6* are shown.

**Figure supplement 7.** ATM-dependent expression of mTOR and AKT signaling is not unique to leukemic cells.

**Figure supplement 8.** Basal and maximal respiration are significantly downregulated with sh*ATM*.

*supplement 5*), further validating our findings. In addition, we confirmed that in other AML cell lines MV4-11 (FLT3-ITD), THP-1 or OCI-AML3 (both FLT3 WT), the degree of ATM knockdown correlated with the extent of reductions in mTOR expression and with AKT phosphorylation in MV4-11 and THP-1 (*Figure 7—figure supplement 6*; *Supplementary file 1*). Furthermore, our analysis of data sets from Gene Expression Omnibus (GEO) indicates that brain tissues from human ataxia-telangiectasia (A-T) patients, who have mutated ATM, have significantly lower *MTOR* mRNA levels compared to control groups, coinciding with a reduction of gene expression associated with mTORC1 signaling and PI3K-AKT-mTOR pathways according to GSEA (*Figure 7—figure supplement 7*). These data implicate that the role of ATM in controlling mTOR signaling extends to other cell types.

We next measured OCR in FLT3-ITD AML cells expressing shRNAs (shNT or sh*ATM*) following inhibition of FLT3 with or without hBMSC-CM. ATM knockdown resulted in a significant reduction of baseline OCR (*Figure 7D*, *Figure 7—figure supplement 8*). More importantly, ATM knockdown in combination with quizartinib prevented the partial restoration of OCR elicited by hBMSC-CM, consistent with our findings with dual inhibition of FLT3 and mTOR. Taken all together, our data indicate that ATM drives hBMSC-CM-mediated protection of AML cells from apoptosis following FLT3 inhibition through regulation of mTOR signaling and oxidative phosphorylation.

## Discussion

Results presented here support a key role for a novel pathway mediated by ATM, mTOR, translational control, and oxidative phosphorylation, whereby BM stromal cells protect FLT3-ITD AML cells from therapeutic elimination following FLT3 inhibition (*Figure 7E*). Multiple components of the hBMSC-CM-dependent pathway are essential for AML survival, as substantiated by the loss of protection when each mediator is targeted by pharmacological inhibition or genetic knockdown. Notably, this survival pathway can be effectively blocked at multiple points using FDA-approved drugs, including everolimus and HHT, which should facilitate the translation of these studies into clinical trials for FLT3-ITD AML. In all, these studies reveal the therapeutic potential of combining FLT3 inhibition with therapeutic targeting of mediators of this pathway to improve clinical outcomes for patients with FLT3-ITD AML.

Although numerous clinical trials have investigated the efficacy of FLT3 inhibitors both as mono-therapy and in combination with other therapeutics for patients with FLT3-mutated AML, these trials have thus far not yielded prolonged remissions (*Levis and Perl, 2020*). This clinical challenge indicates that a better understanding of resistance mechanisms is necessary to identify additional targets for effective molecularly targeted AML therapy. The BM microenvironment has been shown to mediate resistance to FLT3 inhibitors via several mechanisms including (1) high levels of FLT3 ligand in the BM microenvironment that leads to persistent activation of FLT3 *Sato et al., 2011*; (2) cytochrome p450 enzyme produced by BMSCs enhances metabolization of FLT3 inhibitors *Chang et al., 2019*; and (3) BMSCs-mediated activation of prosurvival pathways such as MAPK/ERK and/or STAT5 signaling (*Patel et al., 2020*; *Yang et al., 2014*; *Traer et al., 2016*; *Sung et al., 2019*). Additionally, a recent study has shown that the BM microenvironment is responsible for early resistance of FLT3-ITD+ AML cells to FLT3 inhibition, followed by clonal expansion of pre-existing NRAS mutant subclones that drive relapse (*Joshi et al., 2021*). Collectively, these studies suggest that BM-mediated resistance against FLT3-targeted therapy in FLT3-dependent AML cells is multifactorial and context-dependent.

In our present studies, we focused on intrinsic changes in AML cells upon FLT3 inhibition in the presence of BM microenvironmental factors by utilizing FLT3-ITD AML cell lines and xenograft mouse models utilizing primary FLT3-ITD+ AML cells. Both our in vitro and in vivo data consistently support the conclusion that persistent activation of mTOR signaling mediated by the BM microenvironment impedes cell death induced by FLT3 inhibition. Notably, activation of mTOR signaling has been previously shown in FLT3-WT AML cells co-cultured with mouse BM stroma (*Zeng et al., 2012*), suggesting upregulation of the mTOR pathway in AML cells may not be limited to dependence on FLT3 or human BM stromal factors. Moreover, our therapeutic studies in mice demonstrated the limitation of FLT3-targeted monotherapy in eliminating leukemia from the BM compared to the periphery (spleen and blood), leading to relapse even with sustained treatment. Our results indicate that such relapse can be effectively prevented by combination therapy that targets FLT3 and mTOR signaling. Given that everolimus in combination with midostaurin (a multi-kinase inhibitor that targets FLT3) was well tolerated in patients with AML in a phase 1 clinical study (*Stone et al., 2012*), our results suggest a promising treatment strategy for patients with FLT3-dependent AML.

Over the past two decades, many aspects of mTOR's role in controlling key cellular processes, including protein synthesis, have been extensively elucidated, and mechanistic details of mTOR-mediated pathways in different cellular and physiological contexts remain an area of active study (*Liu and Sabatini, 2020*). Whereas mTOR signaling has long been known to control global protein synthesis, studies have shown that it can also drive the selective translation of specific subsets of mRNAs. Selective translation by mTOR can be determined by the presence of 5'terminal oligopyrimidine (5'TOP) motifs within mRNAs (*Thoreen et al., 2012*). Another study showed that mTOR-dependent selective translation of mRNAs that are important in cell proliferation, survival, and genomic reprogramming confers tamoxifen resistance and estrogen independence in breast cancer cells (*Geter et al., 2017*). Thus, the mechanism of mTOR-driven selective translation is complex and context-dependent, supporting the need for further investigation. Our data reveal a novel process through which mTOR-dependent translational control of transcripts involved in oxidative phosphorylation is a mechanism by which BMSCs mediate survival of FLT3-ITD AML cells following FLT3 inhibition (*Figure 7E*). The mechanism we propose consolidates some of the major findings from studies that explore the role of mTOR signaling and oxidative phosphorylation in different physiological contexts: (1) enhanced mTORC1 signaling is required for translation of mitochondria-related mRNAs during erythropoiesis

(*Liu et al., 2017*); (2) mTORC1 controls mitochondrial respiration and biogenesis via regulation of 4E-BPs in human breast cancer cells (*Morita et al., 2013*); and (3) upregulation of oxidative phosphorylation confers chemoresistance via inhibition of AMPK pathway and subsequent activation of mTORC1 signaling in FLT3-WT AML cells co-cultured with hBMSC (*You et al., 2022*).

Recent studies have shown that drug-resistant AML cells and leukemia stem cells are highly dependent on oxidative phosphorylation for energy production, leading to clinical trials of several drugs that target mitochondrial metabolism, thus directly/indirectly inhibiting oxidative phosphorylation (*Jones et al., 2021*; *de Beauchamp et al., 2022*). Compared to normal hematopoietic cells or solid tumors, AML cells have been shown to have low reserve respiratory capacity, rendering them highly sensitive to inhibition of oxidative phosphorylation and oxidative stress (*Sriskanthadevan et al., 2015*). Our data suggest that the significant restoration of maximal respiration rate and reserve respiratory capacity by hBMSC-CM may promote cell survival by preventing ATP depletion following FLT3 inhibition. While combination therapy effectively reversed the hBMSC-CM-mediated restoration of oxidative phosphorylation and ATP, mTOR inhibition alone did not significantly affect oxidative phosphorylation and ATP levels, suggesting that FLT3-ITD signaling may activate alternative pathway(s) to maintain mitochondrial bioenergetics. Given that low reserve respiratory capacity is a specific metabolic vulnerability of AML cells, we speculate that therapeutic targeting of the mTOR/translation/oxidative phosphorylation axis could block BM-mediated resistance to other AML therapies.

One of the novel aspects of our findings is the unconventional role of ATM in regulation of mTOR signaling and thus oxidative phosphorylation. While many regulators of the mTOR pathway have been identified in a wide range of cellular contexts, the control of the kinase activity of the two major mTOR complexes, mTORC1 and mTORC2, remains a primary focus to understand regulatory inputs of mTOR signaling (*Saxton and Sabatini, 2017*; *Liu and Sabatini, 2020*). Yet, the regulation of mTOR protein levels is poorly understood. We discovered that ATM is required for the expression of the mTOR protein, independent of FLT3 activity or BM stromal effects. It is important to note that ATM-mediated regulation of mTOR protein levels does not appear to be unique to FLT3-dependent AMLs or leukemia in general, as implicated with our analyses of other cell types. Furthermore, our data suggest that ATM not only regulates mTOR protein levels, but also mediates the phosphorylation/activity of AKT. As ATM knockdown only reduced AKT activity in the context of FLT3 inhibition, ATM and FLT3 appear to redundantly regulate AKT activity. Given that AKT has been identified as a downstream effector of FLT3 signaling (*Fathi and Chen, 2011*; *Gilliland and Griffin, 2002*), BM stromal factors likely confer dependence of FLT3-ITD AML cells on ATM following FLT3 inhibition to maintain AKT activity, followed by subsequent activation of mTORC1. Similarly, our observation that S6 and 4E-BP1 were still phosphorylated under baseline condition even in the absence of mTOR in ATM deficient cells suggests that other determinants of S6 and 4E-BP1 activity beyond mTOR exist, perhaps involving other FLT3-dependent signaling molecules.

Based on these intriguing findings, it will be of interest to elucidate the mechanism of ATM-mediated regulation of mTOR signaling. Indeed, ATM has been shown to mediate numerous cellular homeostasis pathways, independent of its functions in response to DNA damage. ATM has been shown to be involved in insulin signaling by promoting protein translation by phosphorylating 4E-BP1 in HEK293 cells and adipose tissue (*Yang and Kastan, 2000*), and by activating AKT via its kinase function to enhance glucose uptake in muscle cells (*Halaby et al., 2008*). In contrast to what we show here, ATM has been shown to *inhibit* mTOR signaling in response to oxidative stress (*Alexander et al., 2010*) or hypoxia (*Cam et al., 2010*), implicating diverse and context-dependent functions of ATM.

In summary, our findings demonstrate that the BM microenvironment provides a unique survival pathway mediated by ATM, mTOR, and oxidative phosphorylation in FLT3-ITD AML cells following FLT3 inhibition. Therapeutic targeting of this pathway, such as by inhibition of mTOR, in combination with FLT3 inhibition could lead to more durable remissions for patients with FLT3-ITD+AML.

# Materials and methods
## Cell lines and cell culture
All human AML cells, MOLM-13 (RRID:CVCL_2119), MV4-11 (CVCL_0064), THP-1 (CVCL_0006), and OCI-AML3 (CVCL_1844) were cultured in RPMI medium supplemented with 10% FBS and 1% antibacterial antimycotic (anti-anti). Other non-AML cells, SUP-B15 (CVCL_0103), were cultured

in IMDM medium supplemented with 10% FBS and 1% anti-anti. The adherent cell line 293FT was grown in DMEM with 10% FBS and 1% anti-anti. All the cell lines were authenticated by short tandem repeat examination and tested negative for Mycoplasma using the e-Myco plus PCR Detection Kit (iNtRON) in October 2021. All transfections were performed in 293FT cells using FuGENE HD Transfection Reagent (#E2311; Promega) at a 3:1:1:1 ratio of shRNA (shNT: #SHC016; or sh$ATM$: #TRCN0000039948 in pLKO.1-puro; MilliporeSigma) or CRISPR-Cas9:sgRNA constructs (sgNT:GAGCTGGACGGCGACGTAAA; or sgATM: TCTACCCCAACAGCGACATG in lentiCRISPR V2) and packaging plasmids pMDLg/pRRE, pMD.G, pRSV-Rev in OPTI-MEM solution (#31985070; Gibco). Viral supernatant was collected 36 and 48 hr after transfection. Spin infections were performed at room temperature at 1500×$g$ for 1 hr with polybrene reagent (1:2000; Fisher Scientific). For the generation of hBMSC-CM, HS-5 cells were cultured in RPMI medium supplemented with 10% FBS and 1% anti-anti. At 80–90% confluency, conditioned media was collected and filtered through 0.22 μm PES sterile filter. hBMSC-CM was prepared by combining 50% of conditioned media of HS-5 with 50% RPMI medium supplemented with 10% FBS and 1% anti-anti. To generate fractionated hBMSC-CM, regular RPMI media was supplemented with heavy fraction of hBMSC-CM isolated by centrifugation of 50% volume of hBMSC-CM using 30 kDa centrifugal filter units (MilliporeSigma).

## Isolation of CD34+ cells

Peripheral blood mononuclear cells (PBMCs) were isolated from fresh human cord blood cells by density centrifugation on Ficoll-Paque, from which CD34+ cells were isolated by the MACS CD34 MicroBead kit (#130-046-702; Miltenyi Biotec). CD34+ cells were incubated in serum-free media (IMDM supplemented with 20% BIT 9500 Serum Substitute, 10 ng/mL SCF, 10 ng/mL IL-3, and 10 ng/mL FLT-3) overnight prior to drug treatments.

## Apoptosis and cell viability assays

Cells were seeded at 1.0–2.0 × 10$^5$/mL in triplicate wells of 48-well tissue culture plates. Where indicated, the cells were treated with drug(s) for 48 hr. After treatment, cells were stained using 7-aminoactinomycin D (7-AAD)/anti-Annexin V (Nexin reagent, MilliporeSigma) to detect apoptotic cells. For cell viability assay, a sample of cells from each well was reseeded at 40,000/mL in RPMI media and incubated for 5 days to measure rebound growth post-treatment. Cells were stained with propidium iodide (PI; 10 mg/mL) and viable cells (PI$^-$) were counted with a flow cytometer (Millipore Guava easyCyte 8HT). Adenosine triphosphate (ATP) levels were measured using the Cell Titer-Glo Assay (Promega). ATP data were normalized to cell number.

## Translation activity measurements

### OPP assay

Cells were treated with the indicated conditions and labeled with 20 μM OPP for 30 min. Protein synthesis was measured using Click-iT Plus OPP Alexa Fluor 488 Protein Synthesis Assay Kit (#C10456; Life Technologies) as per the manufacturer's instructions. Translation activity was quantified by measuring green fluorescence intensity with a flow cytometer (Millipore Guava easyCyte 8HT).

### Polysome profiling

Prior to harvest, cells were treated with the indicated conditions, followed by treatment with 100 μg/mL cycloheximide for 10 min to halt translation. 3 × 10$^7$ cells were harvested from each treatment group. Following harvesting, cells were resuspended in lysis buffer (20 mM HEPES pH 7.4, 15 mM MgCl$_2$, 200 mM NaCl, 1% Triton X-100, 0.1 mg/mL cycloheximide, 2 mM DTT, 100 U/mL RNasin ribonuclease inhibitor). Cell homogenates were spun down for 5 min at 13,000×$g$ to pellet any cellular debris, then 500 μL of this clarified lysate was loaded on 10–60% sucrose gradients in SW41 tubes in lysis buffer lacking Triton X-100. These gradients were prepared using a BioComp system and chilled to 4°C before use. Samples were ultracentrifuged at 36,000 rpm for 3 hr, 10 min, at 4°C, then samples were fractionated using a BioComp system, monitoring absorbance at 260 nm while collecting fractions of approximately 0.4 mL each.

## Oxidative phosphorylation

Real-time mitochondrial respiration was measured using the Seahorse XFe96 Extracellular Flux Analyzer (Agilent Technologies) and Seahorse XF Cell Mito Stress Test kit as per the manufacturer's instructions. AML cells were treated with the indicated conditions prior to Seahorse assay and plated on Seahorse X96 plates coated with 20 µL/well of Cell-TAK adhesive at 25 µg/mL in 0.1 M NaHCO$_3$ at a concentration of $3 \times 10^5$ cells/well suspended in XF base media supplemented with 1 mM pyruvate, 2 mM L-glutamine, 10 mM glucose. OCR measurements were recorded with port injection of oligomycin A (1.5 µM), FCCP (1 µM), and rotenone/antimycin A (0.5 µM) in order. Basal respiration and maximum respiration were calculated as follows. Basal respiration = (Last rate measurement before oligomycin A injection) – (Non-mitochondrial respiration rate), Maximal respiration = (Maximum rate measurement after FCCP injection) –(Non-mitochondrial respiration rate).

## EdU incorporation

10 µM EdU was added to the cells 1 hr before harvest. EdU assays were performed using Click-iT EdU Alexa Fluor 488 Flow Cytometry Assay Kit (#C10420; Life Technologies) as per the manufacturer's instructions. Cell cycle profiles were determined by flow cytometry (Millipore Guava easyCyte 8HT).

## Western blots

Western blot analyses were performed as per the manufacturer's instructions using the following antibodies: ATM (A1106; Sigma-Aldrich), p-ATM S1981 (ab91292; Abcam), cleaved caspase 3 (ab32042; Abcam), mTOR(#2972 and #2983; Cell Signaling), p-mTOR S2448 (#2971; Cell Signaling), c-MYC (#5605; Cell Signaling), p-H2AX S139 (#9718; Cell Signaling), p-4E-BP1 T37/47 (#2855; Cell Signaling), p-4E-BP1 S65 (#9456; Cell Signaling), p-S6 S240/244 (#2215; Cell Signaling), H3(#9715; Cell Signaling). p-AKT S473 (#4060; Cell Signaling), and AKT (#9272; Cell Signaling). Each blot was quantified via densitometry by using Image Lab Software (Bio-Rad). Quantification data is shown in *Supplementary file 1*.

## Therapeutic modeling in mice

NOD.Cg-Rag1[tm1Mom]Il2rg[tm1Wj]/Tg(CMV-IL3,CSF2,KITLG)1Eav/J (NRG-S) mice (Jackson Laboratory, #024099) were bred in-house. The patient samples for xenografting (from Dr. Daniel Pollyea, University of Colorado, Aurora, CO) were obtained from a 69-year-old woman who was newly diagnosed with AML harboring FLT3-ITD and NPM1 mutations (AML #1, sample ID: HTB0097) and a 37-year-old woman who developed relapse AML after chemotherapy (daunorubicin + cytarabine) with FLT3-ITD mutation (AML #2, sample ID: AML100510). Following expansion in vivo, as previously reported (*Alvarez-Calderon et al., 2015*), the secondary leukemia was subsequently used for experiments. Then, 24 hr prior to leukemic transplantation, 6–10-week-old female NRG-S mice were conditioned with 25 mg/kg busulfan delivered by i.p injection. Leukemia cells ($1 \times 10^6$) were injected i.v., and treatment started when peripheral blast counts were between 2% and 5% (mean 3.8%). Leukemic burden was monitored by flow cytometry staining for human HLA-ABC-PE-Cy7 and CD45-FITC (eBioscience). Quizartinib was synthesized and prepared as previously described (*Alvarez-Calderon et al., 2015*), and was dissolved in 30% polyethylene glycol-400, 5% Tween 80 in ddH$_2$O. 2.5 mg/kg or 5 mg/kg quizartinib was delivered once daily p.o. Everolimus was purchased from LC Laboratories and was dissolved in the same solvent as quizartinib. Then, 10 mg/kg everolimus was delivered once daily p.o. After treatments, mice were sacrificed, followed by collection of leukemia cells harvested from the blood, spleen, and BM. Leukemic burden was measured by counting percentages of HLA-ABC+/CD45+ via flow cytometry. For RNA-seq analysis, human leukemia cells were isolated using EasySep Mouse/Human Chimera Isolation Kit (#19849; Stemcell Technologies). All animal experiments were approved by and performed in accordance with guidelines of the Institutional Animal Care and Use Committee at University of Colorado (protocol no. 00170). Deidentified primary AML studies were obtained with donor consent from patients at the University of Colorado Anschutz Medical Campus (COMIRB protocol #12-0173).

### Reverse-phase protein array (RPPA)

MOLM-13 cells were treated with vehicle or 3 nM quizartinib in either RPMI or hBMSC-CM for 16 hr, followed by washing with PBS. Cell pellets were frozen and sent to the RPPA Core Facility at the University of Texas MD Anderson Cancer Center.

### RNA-seq

Following drug treatments and cell isolation from in vitro and in vivo studies as described above, RNA from each sample was isolated using RNeasy Plus Mini Kit (#73134; QIAGEN). Poly-A-selected total RNA library construction was performed using Universal Plus mRNA-seq with NuQuant (#0520-A01; Tecan), and paired-end sequencing was performed on a NovaSeq6000 instrument (Illumina) in the Genomics Shared Resource (University of Colorado). Illumina adapters were trimmed and reads <50 base pairs were removed with BBDuk (*Bushnell, 2019*). Trimmed reads were aligned to the human Ensembl genome (hg38.p12, release 96) and gene counts were quantified using STAR v2.6.0a (*Dobin et al., 2013*). Ensembl IDs were mapped to gene names and counts of genes with multiple IDs were aggregated. Lowly expressed genes were defined as <1 mean raw count or <1 count per million across the dataset and were removed from the analysis. The limma R package (*Ritchie et al., 2015*) was used to calculate differential expression between the indicated groups. An interaction model was used with the input fractions for the polysome comparisons. Pathway analysis was performed using fold-change and the fgsea R package (*Sergushichev, 2016*), and Hallmark or GO terms from the Molecular Signatures Database (*Liberzon et al., 2011*), which were downloaded using the msigdbr R package (*Dolgalev, 2021*). Heatmaps were generated using the complexheatmap R package *Gu et al., 2016* following counts per million normalization and z-score transformation. Raw and processed data has been deposited to the GEO. GEO identifier is GSE202230.

### Public analysis of microarray data

Normalized expression was downloaded directly from GEO (GSE61019). AT case samples were compared to control using limma, and pathway analysis was performed as described above.

### Statistical analysis

Data are represented as box plots showing median values, with the boundaries of the rectangle representing the first and third quartiles, while whiskers extend from the minimum to maximum points in each plot. The number of replicates (n) is reported in the figure legends. Comparisons between two values were performed by Student's t-test (unpaired two-tailed). Significance was defined at $*p \leq 0.05$, $**p \leq 0.01$, $***p \leq 0.001$, $****p \leq 0.0001$.

## Acknowledgements

This work was supported by grants from NCI NRSA F30CA23197 (to HJP), R35GM118070 (to JSK), the V Foundation (T2016-012) and St. Baldrick's Foundation (AWD-430131) (to JDG), and the Leukemia and Lymphoma Society (7020-19) (to CTJ and JDG). The authors thank Dr. Daniel Pollyea for providing human primary samples, Dr. Brett Stevens for critical reading of the manuscript, and the Bioinformatics and Biostatistics, Genomics and Functional Genomics Shared Resources supported by National Cancer Institute grant P30CA046934 to the University of Colorado Cancer Center. The authors thank the RPPA Core Facility at the University of Texas MD Anderson Cancer Center (NCI CA16672 and NIH R50CA22165).

## Additional information

### Funding

| Funder | Grant reference number | Author |
|---|---|---|
| National Cancer Institute | Graduate Student Fellowship NRSA F30CA23197 | Hae J Park |

| Funder | Grant reference number | Author |
| --- | --- | --- |
| National Cancer Institute | R35GM118070 | Jeffrey S Kieft |
| V Foundation for Cancer Research | T2016-012 | James DeGregori |
| St. Baldrick's Foundation | AWD-430131 | James DeGregori |
| Leukemia and Lymphoma Society | 7020-19 | Craig T Jordan James DeGregori |
| National Cancer Institute | P30CA046934 | James DeGregori |

The funders had no role in study design, data collection and interpretation, or the decision to submit the work for publication.

## Author contributions

Hae J Park, Conceptualization, Data curation, Formal analysis, Validation, Investigation, Visualization, Methodology, Writing - original draft, Writing - review and editing; Mark A Gregory, Conceptualization, Validation, Investigation, Methodology, Writing - review and editing; Vadym Zaberezhnyy, Data curation, Investigation; Andrew Goodspeed, Formal analysis; Craig T Jordan, Conceptualization, Resources, Writing - review and editing; Jeffrey S Kieft, Data curation, Investigation, Methodology; James DeGregori, Conceptualization, Resources, Supervision, Funding acquisition, Validation, Investigation, Visualization, Methodology, Project administration, Writing - review and editing

## Author ORCIDs

Hae J Park (iD) http://orcid.org/0000-0001-7768-9297
James DeGregori (iD) http://orcid.org/0000-0002-1287-1976

## Ethics

Deidentified primary AML studies were obtained with donor consent from patients at the University of Colorado Anschutz Medical Campus (COMIRB protocol #12-0173).
All animal experiments were approved by and performed in accordance with guidelines of the Institutional Animal Care and Use Committee at University of Colorado (protocol No. 00170).

## Decision letter and Author response

Decision letter https://doi.org/10.7554/eLife.79940.sa1
Author response https://doi.org/10.7554/eLife.79940.sa2

# Additional files

## Supplementary files

• Supplementary file 1. Each western blot was quantified via densitometry by using Image Lab Software (Bio-Rad). Quantification data with statistical analyses including mean and standard deviation from each experiment (n = 2–3) is shown.

• MDAR checklist

## Data availability

All data generated or analyzed during this study are included in manuscript and supporting file. RNA-seq data have been deposited in GEO under accession codes GSE202230.

The following dataset was generated:

| Author(s) | Year | Dataset title | Dataset URL | Database and Identifier |
| --- | --- | --- | --- | --- |
| Park H | 2022 | Investigation of bone marrow mediated therapeutic resistance to FLT3 inhibition in acute myeloid leukemia cells | https://www.ncbi.nlm.nih.gov/geo/query/acc.cgi?acc=GSE202230 | NCBI Gene Expression Omnibus, GSE202230 |

The following previously published dataset was used:

| Author(s) | Year | Dataset title | Dataset URL | Database and Identifier |
|---|---|---|---|---|
| Jiang D, Zhang Y, Hart RP, Chen J | 2015 | Ataxia-telangiectasia cerebellar cortex expression | https://www.ncbi.nlm.nih.gov/geo/query/acc.cgi?acc=GSE61019 | NCBI Gene Expression Omnibus, GSE61019 |

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
