## [Editor Report]

FLT3 (Fms-related receptor tyrosine kinase 3) activation occurs in a subset of AML cases and is associated with poor prognosis. This work is focused on the mechanisms of resistance to FLT3 inhibitors in AML. The authors show that the combination of the FLT3 inhibitor and an mTORC1 inhibitor reduces tumor burden and prevents relapse in FLT3 mutant AML. This article is of interest to scientists and physicians investigating AML as well as scientists studying signaling pathways.

---

## [Decision Letter]

**Decision letter after peer review:**

Thank you for submitting your article "Therapeutic resistance in acute myeloid leukemia cells is mediated by a novel ATM/mTOR pathway regulating oxidative phosphorylation" for consideration by *eLife*. Your article has been reviewed by 3 peer reviewers, one of whom is a member of our Board of Reviewing Editors, and the evaluation has been overseen by W Kimryn Rathmell as the Senior Editor. The reviewers have opted to remain anonymous.

The reviewers have discussed their reviews with one another, and the Reviewing Editor has drafted this to help you prepare a revised submission. Please note, that only a manuscript that has undergone substantial revision will be considered.

Essential revisions:

1) Include MV411 cells in the mechanistic experiments.

2) Revise the conclusion (and title of Figure 2) that the CM "reverses mTOR pathway suppression" Also for Figure 2 – These experiments should be done in triplicate, with statistics, and other indicators of mTOR activity included (pS6K, p4EBP1) along with the total protein levels for all of the phospho-proteins including S6. These experiments should also be done in the MV411 cells.

3) In Figure S3 (AML#2), everolimus alone nearly completely suppressed the leukemic cells in the blood, consistent with the possibility that it has actions that are unrelated to Quizartinib. The authors should address this.

4) In Figure 3G, (AML#1) include no vehicle data for the 21 Day treatment time point.

What was the leukemia burden in the vehicle? Without this, the data are hard to interpret. It appears that the combination is not different from Quiz alone.

5) Revise the title of Figure 4 "Restoration of mTOR signaling by hBMSC-CM …" to reflect the data better.

6) Figure 4 and elsewhere include MV411 cells.

7) Are the changes consistent among the different experimental settings? If this is the case, are these changes in gene expression consistent with PGC1s or ERRs OXPHOS target genes that are controlled via mTOR, see Cunningham et al. Nature 2007 Nov 29;450(7170):736-40 or Dufour et al. Cell Rep. 2022 Mar 22;38(12):110534.

8) The authors used hBMSC conditioned medium to mimic the bone marrow environment and used normal RPMI media as a control, to investigate the effects of bone marrow-released stromal factors on the TLT3 inhibitor resistance of AML. However, in addition to the "stromal factors", conditioned medium has different pH, metabolites, glucose levels, and more compared to RPMI medium, making the system complicated. Can the authors use the conditioned medium from HEK-293 or a spleen cell line as the control?

9) Explain why the treatment times of drugs changed from experiment to experiment. Is there any reason for this?

10) Improve the Western blot quality throughout.

11) Explain why in Figure 7C, knockdown of ATM completely eliminates mTOR but the translation indicators, p-4EBP1, and p-S6 are kept unchanged, indicating inhibition of mTOR in these cells has no effect on protein translation in general. This appears to contradict the data shown previously in this manuscript.

12) better place this work in the context of prior work. Prior papers report that the combination of FLT3 and mTOR inhibitors are synergistic, although they don't discuss stromal protection.

13) Address the issue of FLT3 inhibitor concentration. The experiments utilize a single concentration of FLT3 inhibitors. It would be helpful to understand the effects of a range of concentrations. Are the protective effects of stroma lost at higher concentrations? Do they see synergy without stroma at different concentrations?

14) It is interesting that polysome fractions show enrichment of oxphos transcripts, but gene expression pathways are not different. Can the authors speculate on how the polysome fractions of oxphos genes are increasing?

*Reviewer #3 (Recommendations for the authors):*

1) Prior papers report that the combination of FLT3 and mTOR inhibitors are synergistic, although they don't discuss stromal protection. Thus, some of the novelty of this paper is reduced. Can the authors place their work in the context of the published data.

2) The experiments utilize a single concentration of FLT3 inhibitors. It would be helpful to understand the effects of a range of concentrations. Is the protective effects of stroma lost at higher concentrations? Do they see synergy without stroma at different concentrations?

3) An important area of novelty is that stroma maintains/increases mTOR signaling. However, more mechanistic studies would be needed to link the stroma to mTOR.

4) It is interesting that polysome fractions show enrichment of oxphos transcripts, but gene expression pathways are not different. Can the authors speculate on how the polysome fractions of oxphos genes are increasing?

5) Are similar changes seen in AML patients treated with FLT3 inhibitors?

---

## [Author Response]

Essential revisions:1) Include MV411 cells in the mechanistic experiments.

We generated data demonstrating that MV4-11 cells show similar phenotypes as MOLM-13 cells in multiple mechanistic experiments. These experiments include: (1) time course western blot where hBMSC-CM significantly prevented downregulation of mTOR activity as well as c-MYC expression upon FLT3 inhibition in MV4-11 cells (Figure 2—figure supplement 1); (2) OPP assay in which MV4-11 cells showed mTOR-dependent partial restoration of translation activity mediated by hBMSC-CM upon FLT3 inhibition (Figure 4—figure supplement 2); (3) Seahorse assays in which MV4-11 cells demonstrated mTOR-dependent partial restoration of OCR following FLT3 inhibition in the presence of hBMSC-CM (Figure 6—figure supplement 1A); (4) ATP assays that show ATP levels corresponding to changes in OCR (Figure 6—figure supplement 1B); (5) apoptosis assays in which shRNA-mediated knockdown of ATM in MV4-11 cells reversed hBMSC-CM-mediated protection from apoptosis following FLT3 inhibition (Figure 7—figure supplement 3).

2) Revise the conclusion (and title of Figure 2) that the CM "reverses mTOR pathway suppression" Also for Figure 2 – These experiments should be done in triplicate, with statistics, and other indicators of mTOR activity included (pS6K, p4EBP1) along with the total protein levels for all of the phospho-proteins including S6. These experiments should also be done in the MV411 cells.

We revised the conclusion and title of Figure 2 as “Bone marrow microenvironment substantially limits downregulation of mTOR and MYC pathway in FLT3-ITD AML cells upon FLT3 inhibition in vitro and in vivo.” In addition, we performed time course western blot experiments (Figure 2D) in biological triplicate and revised Supplementary File 1 to include quantification of each experiment analyzed by western blotting including standard deviation. We improved the western blot quality of Figure 2D and included total protein levels of mTOR, S6 and 4E-BP1. We also included the time course western blot for MV4-11 cells in Figure 2—figure supplement 1, with quantification in the Supplementary File 1.

3) In Figure S3 (AML#2), everolimus alone nearly completely suppressed the leukemic cells in the blood, consistent with the possibility that it has actions that are unrelated to Quizartinib. The authors should address this.

We addressed this concern on page 7 as follow:

“Notably, everolimus alone demonstrated similar effectiveness as quizartinib monotherapy in eliminating leukemia cells only in the blood without showing significant effect in the spleen or bone marrow, suggesting that mTOR inhibition alone results in different cell killing effects depending upon tissue microenvironment”

4) In Figure 3G, (AML#1) include no vehicle data for the 21 Day treatment time point.What was the leukemia burden in the vehicle? Without this, the data are hard to interpret. It appears that the combination is not different from Quiz alone.

Figure 3G shows results from mice harvested 84 days after treatment initiation. The vehicle treated mice had a median survival of 27 days, and all had succumbed by 84 days. Thus, we could not include a vehicle group for this late timepoint, as treatment is required for survival to this point. We have clarified this in the text. We also note that the combination therapy is substantially and significantly more effective than quizartinib alone in eliminating leukemic burden from all analyzed tissues.

5) Revise the title of Figure 4 "Restoration of mTOR signaling by hBMSC-CM …" to reflect the data better.

We changed the title of Figure 4 as “Suppression of protein translation upon FLT3 inhibition is significantly alleviated by hBMSC-CM via an mTOR-dependent mechanism”

6) Figure 4 and elsewhere include MV411 cells.

As described in #1, we included OPP assay results of MV4-11 cells in Figure 4—figure supplement 2. These results are now discussed on page 8.

7) Are the changes consistent among the different experimental settings? If this is the case, are these changes in gene expression consistent with PGC1s or ERRs OXPHOS target genes that are controlled via mTOR, see Cunningham et al. Nature 2007 Nov 29;450(7170):736-40 or Dufour et al. Cell Rep. 2022 Mar 22;38(12):110534.

We thank the reviewer for this suggestion, which led to some interesting results, now discussed on page 10:

“Given that mTOR has been shown to control transcription of mitochondrial genes through a complex involving transcription factors such as PGC1-α and YY1 (33), we further compared levels of transcripts in input samples of multiple mitochondrial genes and transcription factors that regulate the expression of mitochondrial genes. Interestingly, PGC1-α and the mitochondrial enzyme IDH3A demonstrated mTOR-dependent restoration by hBMSC-CM following FLT3 inhibition (Figure 5—figure supplement 1). PGC1-α is a transcriptional coactivator, and thus its mTOR dependent restoration by hBMSC-CM could contribute to the partial restoration of OXPHOS, a possibility that would require further investigation. Notably, other mitochondrial transcriptional regulators and mitochondrial genes examined did not show such patterns.”

8) The authors used hBMSC conditioned medium to mimic the bone marrow environment and used normal RPMI media as a control, to investigate the effects of bone marrow-released stromal factors on the TLT3 inhibitor resistance of AML. However, in addition to the "stromal factors", conditioned medium has different pH, metabolites, glucose levels, and more compared to RPMI medium, making the system complicated. Can the authors use the conditioned medium from HEK-293 or a spleen cell line as the control?

We addressed this concern on page 4 as follows:

“Given that hBMSC-CM could contain different levels of metabolites and nutrients compared to regular RPMI media, we further tested if FLT3-ITD AML cells respond differently to quizartinib in regular RPMI supplemented with fractionated hBMSC-CM. RPMI supplemented with the heavy fraction of hBMSC-CM that contains molecules larger than 30kDa phenocopied the protective effect of hBMSC-CM, while the light fraction of hBMSC-CM containing molecules smaller than 30kDa demonstrated complete loss of protection from apoptosis upon FLT3 inhibition, suggesting that the protective effects of hBMSC-CM are due to larger molecules and cannot be explained by altered levels of smaller metabolites and nutrients (Figure 1D). Moreover, conditioned media from HS-27A, another human BM stromal cell line, did not provide protection from apoptosis upon FLT3 inhibition (Figure 1—figure supplement 3). Together with the previous finding that only HS-5 cells recapitulated the general expression pattern of primary BM mesenchymal stromal cells while HS-27A cells displayed a distinctive gene expression profile (19), our data indicate that protection may require specific soluble factor(s). To understand key soluble factors responsible for hBMSC-CM-mediated protection, we examined if cytokines enriched in BM microenvironment such as GM-CSF, IL-3 and FGF-2 that have been shown to protect FLT3-ITD AML cells from cell killing by FLT3 inhibitors (17, 18) show similar effects to hBMSC-CM. While GM-CSF or/and IL-3 induced modest and partial protection from apoptosis upon FLT3 inhibition, FGF-2 did not result in significant protection (Figure 1—figure supplement 4), indicating that hBMSC-CM-mediated survival of FLT3-ITD AML cells from FLT3 inhibition likely involves multiple factors.”

9) Explain why the treatment times of drugs changed from experiment to experiment. Is there any reason for this?

To determine the cell killing effects of drug treatments, we chose to measure apoptosis at 48 hours post drug treatment, a time point where cell death with the treatments is consistently evident. Following 48 hours of drug treatment, we measured cell viability at 5 days after reseeding cells from each condition to measure rebound growth, which reflects the ability of cells to recover and expand post drug treatments. For measurements of biochemical activities, such as RNA expression, protein levels, translation activity, and oxidative phosphorylation, we used shorter time points from 12-16 hours. Given that apoptosis begins about 20 hours post quizartinib, we chose these earlier timepoints to allow sufficient time for these parameters to change, but late enough to avoid the secondary effects of cell death activation.

10) Improve the Western blot quality throughout.

We repeated multiple Western blots, substantially improved the image quality of Figure 2D (particularly mTOR signaling western per reviewer 2’s concern #4), and improved image quality of other western blot figures.

11) Explain why In Figure 7C, knockdown of ATM completely eliminates mTOR but the translation indicators, p-4EBP1, and p-S6 are kept unchanged, indicating inhibition of mTOR in these cells has no effect on protein translation in general. This appears to contradict the data shown previously in this manuscript.

We have provided an explanation for this result in the Discussion section (on page 15):

“our observation that S6 and 4E-BP1 were still phosphorylated under baseline condition even in the absence of mTOR in ATM deficient cells suggests that other determinants of S6 and 4EBP1 activity beyond mTOR exist, perhaps involving other FLT3-dependent signaling molecules.”

12) better place this work in the context of prior work. Prior papers report that the combination of FLT3 and mTOR inhibitors are synergistic, although they don't discuss stromal protection.

We addressed this concern on page 6 as follows:

“As a downstream effector of activated FLT3 kinase, mTOR signaling has been implicated in the survival of FLT3-ITD^+^ AML cells, and aberrant activation of the mTOR pathway has been reported in FLT3-ITD AML cell lines that developed intrinsic resistance to FLT3 inhibitors through long-term culture with FLT3 inhibitor, coinciding with enhanced sensitivity to the combination of FLT3 and mTOR inhibitors (26,27).”

13) Address the issue of FLT3 inhibitor concentration. The experiments utilize a single concentration of FLT3 inhibitors. It would be helpful to understand the effects of a range of concentrations. Are the protective effects of stroma lost at higher concentrations? Do they see synergy without stroma at different concentrations?

We addressed this concern on page 7 as follows:

“In addition, combinatorial treatment of everolimus with a range of quizartinib concentrations demonstrated significantly greater elimination than quizartinib alone in both regular media RPMI and hBMSC-CM (Figure 3—figure supplement 2), although the extent of this protection by hBMSC-CM was reduced at the highest dose of quizartinib, potentially due to off-target effects.”

14) It is interesting that polysome fractions show enrichment of oxphos transcripts, but gene expression pathways are not different. Can the authors speculate on how the polysome fractions of oxphos genes are increasing?

We elaborated on this result in the revised Discussion section (on page 14):

“Over the past two decades, many aspects of mTOR’s role in controlling key cellular processes, including protein synthesis, have been extensively elucidated, and mechanistic details of mTOR-mediated pathways in different cellular and physiological contexts remain an area of active study (45). Whereas mTOR signaling has long been known to control global protein synthesis, studies have shown that it can also drive the selective translation of specific subsets of mRNAs. Selective translation by mTOR can be determined by the presence of 5’ terminal oligopyrimidine (5’TOP) motifs within mRNAs (46). Another study showed that mTOR-dependent selective translation of mRNAs that are important in cell proliferation, survival, and genomic reprogramming confers tamoxifen resistance and estrogen independence in breast cancer cells (47). Thus, the mechanism of mTOR-driven selective translation is complex and context-dependent, supporting the need for further investigation. Our data reveal a novel process through which mTOR-dependent translational control of transcripts involved in oxidative phosphorylation is a mechanism by which BMSCs mediate survival of FLT3-ITD AML cells following FLT3 inhibition (Figure 7E). The mechanism we propose consolidates some of the major findings from studies that explore the role of mTOR signaling and oxidative phosphorylation in different physiological contexts: (1) Enhanced mTORC1 signaling is required for translation of mitochondria-related mRNAs during erythropoiesis (48); (2) mTORC1 controls mitochondrial respiration and biogenesis via regulation of 4E-BPs in human breast cancer cells (49); (3) Upregulation of oxidative phosphorylation confers chemoresistance via inhibition of AMPK pathway and subsequent activation of mTORC1 signaling in FLT3-WT AML cells co-cultured with hBMSC (50).”

Reviewer #3 (Recommendations for the authors):3) An important area of novelty is that stroma maintains/increases mTOR signaling. However, more mechanistic studies would be needed to link the stroma to mTOR.

We now show that stromal factor(s) greater than 30KDa is essential for survival of AML cells following FLT3 inhibition. Still, we do not yet know what stromal factor(s) mediate the protection of AML cells from apoptosis through the ATM/mTOR/OXPHOS pathway. A new graduate student in the lab is working to identify the mechanistic details that link the BM stroma to ATM/mTOR/OXPHOS – we believe that these efforts will represent a multi-year project.

5) Are similar changes seen in AML patients treated with FLT3 inhibitors?

This is of great interest for us as it would be clinically relevant. We tried to obtain primary AML samples from patients treated with FLT3 inhibitors from our clinical collaborators, but we could not obtain sufficient numbers to allow this question to be addressed, given variability in treatments (e.g. often with standard chemotherapy) and variability of bone marrow sampling (often at points where leukemic burden was very low). We are in discussions with colleagues to translate our discoveries into a clinical trial combining FLT3 and mTOR inhibitors, and characterization of biochemical responses will be part of this trial.